# Sedimentary noise and sea levels linked to land–ocean water exchange and obliquity forcing

Mingsong Li [1,2,3], Linda A. Hinnov[2,1], Chunju Huang[1,4] & James G. Ogg[5,1]

In ancient hothouses lacking ice sheets, the origins of large, million-year (myr)-scale sea-level oscillations remain a mystery, challenging current models of sea-level change. To address this mystery, we develop a sedimentary noise model for sea-level changes that simultaneously estimates geologic time and sea level from astronomically forced marginal marine stratigraphy. The noise model involves two complementary approaches: dynamic noise after orbital tuning (DYNOT) and lag-1 autocorrelation coefficient ($\rho_1$). Noise modeling of Lower Triassic marine slope stratigraphy in South China reveal evidence for global sea-level variations in the Early Triassic hothouse that are anti-phased with continental water storage variations in the Germanic Basin. This supports the hypothesis that long-period (1-2 myr) astronomically forced water mass exchange between land and ocean reservoirs is a missing link for reconciling geological records and models for sea-level change during non-glacial periods.

[1] State Key Laboratory of Biogeology and Environmental Geology, School of Earth Sciences, China University of Geosciences, Wuhan 430074 Hubei, China. [2] Department of Atmospheric, Oceanic, and Earth Sciences, George Mason University, Fairfax, Virginia 22030, USA. [3] Department of Geosciences, Pennsylvania State University, University Park, PA 16802, USA. [4] Laboratory of Critical Zone Evolution, School of Earth Sciences, China University of Geosciences, Wuhan 430074 Hubei, China. [5] Department of Earth, Atmospheric and Planetary Sciences, Purdue University, West Lafayette, IN 47907, USA. Correspondence and requests for materials should be addressed to M.L. (email: limingsonglms@gmail.com) or to C.H. (email: huangcj@cug.edu.cn)

The Earth's stratigraphic record of past sea-level variations provides fundamental insights into the dynamics of present-day global sea-level change. Global sea-level variations result from changes in ocean basin capacity and seawater volume[1]. Ocean basin capacity changes are dominated by low-frequency ($<10^{-6}$ per year) variations in sea-floor spreading rate[2]. Higher frequency ($>10^{-6}$ per year) variations in seawater volume, i.e., eustasy are associated with the astronomically forced growth and decay of continental ice sheets that produce high-amplitude eustatic changes (up to 200 m)[2]. Other processes that pace seawater volume also occur at high-frequency but with low-amplitudes (5–10 m): variations in continental groundwater and lake storage, and thermal expansion and contraction of seawater[3,4]. The causes of million-year (myr) scale, high-amplitude (>75 m) sea-level oscillations under hothouse conditions in the absence of continental ice sheets, e.g., during the early Triassic Period, remain unknown[1,5–7].

**Fig. 1** Synthetic stratigraphic noise model of sea-level variations. **a** High sea-level and wave base with less water depth-related 'noise' at location of observer. **b** Low sea-level and wave base with more depth-related 'noise' and/or gaps at location of observer. **c** $2\pi$ multitaper power spectrum of Laskar2004 nominal astronomical solution (La04ETP, sum of standardized eccentricity (E), obliquity, or tilt (T), and precession (P)) of 0–10 Ma plotted against the united E, T and P bands (blue shading). **d** La04ETP from 0 to 10 Ma. **e** DYNOT model of La04ETP. **f** $\rho_1$ model of La04ETP. **g** Seven intervals with different types of noise (black) and gaps (red): (α) 500 kyr white Gaussian noise, (β) 500 kyr red noise with a $\rho_1 = 0.3$, (γ) multiple intermittent, brief gaps, (δ) 4.75-myr white Gaussian noise with (ε) additional 500 kyr white Gaussian noise, (ζ) a 100 kyr gap, and (η) a 750-kyr gap. **h** Sum of La04ETP in **d** and the noise in **g**. ETP series expanded for 'unrecognized' gaps (γ) or zero during the 'recognized' gap (ζ) in **g** or missing during the 'unrecognized' gap (η). **i, j** DYNOT and $\rho_1$ models of the series in **h**, curve minima correspond to noise in **g** and multiple gaps (γ) are not revealed in **j**. Confidence levels are estimated by a Monte Carlo analysis with 5,000 iterations and a running window of 400 kyr

The geologic history of sea-level has been reconstructed from seawater volume proxies and marginal marine depositional sequences. Different proxies in sedimentary sections lead to diverse interpretations[1,2]. The magnitude of sea-level estimates for the past 100 million years rely heavily on foraminiferal calcite oxygen isotopes ($\delta^{18}O$), which are influenced by temperature, evaporation and precipitation, and diagenesis[2]. Sequence stratigraphy addresses stratal stacking patterns and changes thereof in a chronological framework[8]. Developments in sequence stratigraphy have greatly clarified the origin of genetically related sedimentary packages related to sea-level change, and have facilitated the reconstruction of sea level through geologic time[2,5,6,9]. However, problems in sequence stratigraphy persist with confusing and even conflicting terminology, multiple depositional models, difficulties in recognition and correlation of sequence stratigraphic surfaces, and subjective assessment of sequence hierarchical order[8]. For example, sedimentary features representing sea-level fall in depositional sequences are often marked by unconformable surfaces in basin margins. Toward the basin center these unconformable surfaces may be subtle and even 'conformable' thus difficult or even impossible to identify. These problems together with limited accuracy in the geologic timescale hinder the reconstruction of global sea-level and understanding the origins of sea-level change.

Here we present stratigraphic evidence that elucidates the causes of high frequency, high-amplitude sea-level changes that occurred during the ice-free Early Triassic hothouse. We develop a dynamic noise after orbital tuning, or DYNOT model for the recognition of sea-level variations based on the dynamic non-orbital signal in climate proxy records after subtracting orbital, i.e., astronomically forced climate signal. The DYNOT model is supplemented by a second, independent lag-1 autocorrelation coefficient, or $\rho_1$ model, which forms the basis of a well-established statistical method for red noise estimation of time series[10–12]. DYNOT and $\rho_1$ modeling applied to a marine slope gamma ray record from the past 1.4 myr correlate with sea-level changes reconstructed from benthic foraminiferal $\delta^{18}O$. This verification indicates that the sedimentary noise model is a useful method for sea-level reconstruction. These two approaches for modeling sedimentary noise applied together with an astronomical timescale for the Early Triassic[13,14] enable correlation among time series of global sea-level, continental water storage and astronomical climate forcing.

## Results

**Modeling dynamic sedimentary noise.** Climate and sea-level proxy variations consist of long-term trends, $10^6$ year-scale orbital (eccentricity and inclination) modulation cycles, $10^3$ to $10^5$ year-scale astronomical (orbital eccentricity, obliquity and precession) cycles, $10^0$ to $10^3$ year-scale climate variability, and abrupt geological events. Importantly, an abundance of 'noise' is also embedded in the originating climate signals. Sources of noise that affect climate and sea-level proxies can be classified as follows: water-depth related noise such as storms, tides, bioturbation, and unsteady depositional rate; proxy-related noise including proxy sensitivity, measurement error, non-linear climate response, and dating error[11]; and other factors such as tectonics, volcanism and post-depositional diagenesis[15]. Among these, measurement error and proxy sensitivity can be assessed from replicate proxy data sampled across a single stratigraphic interval, and dating errors and depositional rate by generating an age model, e.g., astrochronology. Variations in the water-depth related noise at a fixed location in the marginal marine environment are related to relative sea-level changes (Fig. 1). When sea level is relatively high, water-depth related noise at a fixed slope

location in the marginal marine environment is weaker than the noise in a time of relatively low sea-level, and vice versa.

**DYNOT sea-level model.** The DYNOT model is designed to measure noise in climate and sea-level proxies. If proxy-related noise and other factors (see above) are minor, the variance of the noise can be an indicator for relative sea-level changes. For a $10^3$ to $10^6$ year-band in the power spectra of proxy series, we evaluate the ratio of non-orbital signal variance to the total variance, which is calculated along a sliding time window (Methods). When sea-level is relatively high, the DYNOT ratio is weaker than the ratio in a time of relatively low sea-level, and vice versa.

**$\rho_1$ sea-level model.** Climate change tends to incorporate previous values over a range of timescales; this is termed autocorrelation or persistence[10,11]. The simplest and most widely adopted persistence model is based on the lag-1 autocorrelation coefficient[10,11,16]. The $\rho_1$ model is tested as a second, independent noise indicator for relative sea-level change (Methods). As demonstrated below, increased noise leads to a decreased $\rho_1$ value, and vice versa.

**Applications and restrictions.** The sedimentary noise model is expected to apply to proxies that are sensitive to water-depth-related noise. In this application, the model is assumed to be valid for slope and basin environments at water depths of several meters to several hundred meters that are near or just below storm-wave base, where storms, tides, bioturbation, and unsteady depositional rate are expected to exert measurable influence (noise) in sedimentary records.

Other factors may contribute to sedimentary noise related to sea-level change. For example, basin-scale tectonic activity may affect $10^6$ year-scale relative sea-level changes[2]. Short-term tectonic activity such as earthquake-induced downslope movements may affect sea-level change by imposing sudden (and random) jumps or spikes leading to elevated noise at all frequencies. Non-linear responses of sea-level change to orbital forcing may generate spectral sidebands or combination tones[17] that cannot be removed by the model. Simulation of non-linear accumulation and bioturbation effects on precession-forced basinal carbonate cycles demonstrate that variance can be transferred from the precession into the eccentricity band[18], generating 'redder', i.e., higher $\rho_1$ values. Sub-Milankovitch scale climate oscillations captured in very high-resolution proxy records will not be removed by DYNOT modeling. Geological events such as volcanism may lead to hydrological changes and proxy records with extended perturbations with increased low-frequency non-orbital noise, thus impacting both approaches. Changes in the Earth's climate state may generate additional noise and affect the climate persistence, contributing to noise in deep ocean records[10]. However, the model verification study presented below does not show evidence of this type of influence in a Quaternary marginal marine environment. Interpolation of irregularly spaced data can also affect the model, for example, upsampling to increase sampling rate leads to artificially high $\rho_1$ values (Methods). Finally, post-depositional diagenesis of calcareous bedding may enhance and/or distort orbital-scale variations. These factors are discussed in detail in the Supplementary Note 1. DYNOT and $\rho_1$ modeling of sedimentary noise is nonetheless powerful for sea-level reconstruction as demonstrated below.

**Testing the sedimentary noise model.** The efficacy of DYNOT and $\rho_1$ modeling is demonstrated on the La2004 astronomical solution[19] from 0 to 10 Ma with added synthetic noise and gaps (Fig. 1). The added noise simulates different environmental

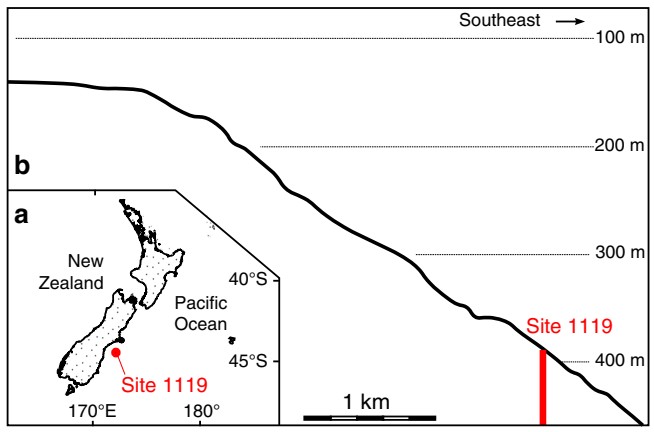

**Fig. 2** Locality maps. **a** Locality map of ODP Site 1119[23]. **b** Profile through Site 1119 showing present-day water depth[20]

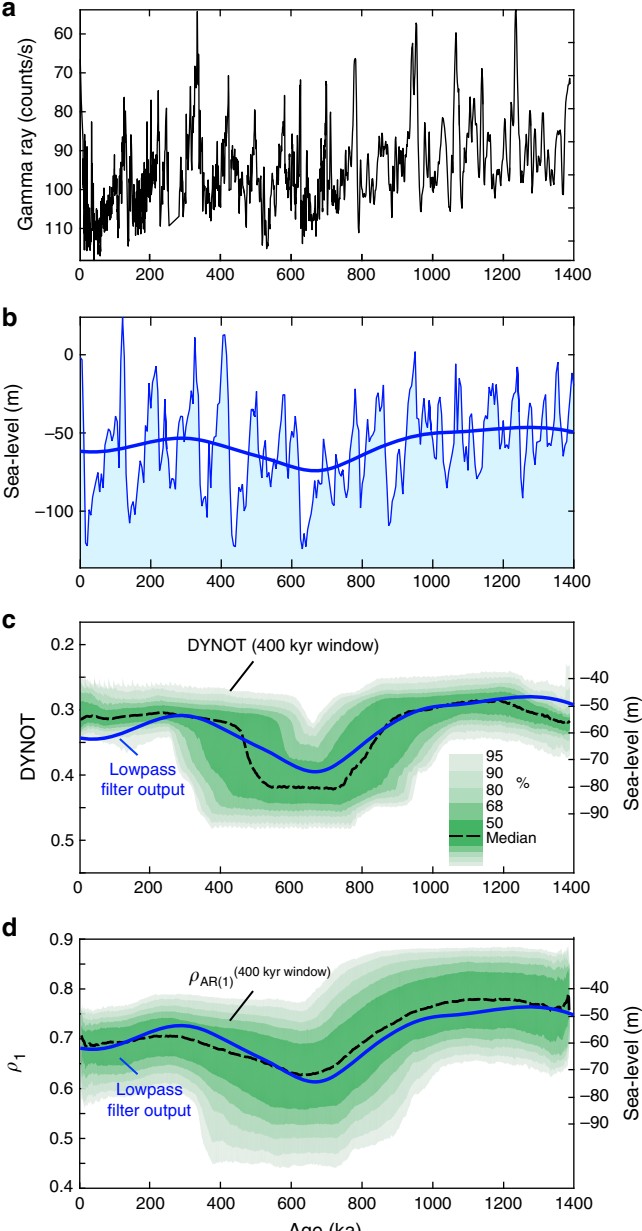

**Fig. 3** Testing sedimentary noise models on the gamma ray log from ODP Site 1119. **a** Fine-tuned gamma ray series, with one outlier removed (see Supplementary Fig. 3). **b** Sea-level changes are estimated from benthic foraminiferal $\delta^{18}O$ [2] shown with lowpass Gaussian filter output of sea-level changes (thick blue; cutoff frequency is 1/(400 kyr) using 'gaussfilter.m[50]'). **c** DYNOT models of interpolated, tuned time series in **a** using random sample rates of 0.22–2.04 kyr (Methods) with a running window of 400 kyr. **d** $\rho_1$ model using random sample rates of 2.04–3.06 kyr (Methods) with a running window of 400 kyr. The DYNOT and $\rho_1$ results are shown with lowpass filter output of sea-level changes in **b** (solid blue). Confidence intervals are estimated by Monte Carlo analysis with 5000 iterations

processes under a variety of sea-level conditions. Strong noise (low sea level) leads to elevated DYNOT and decreased $\rho_1$ values, and vice versa. Intermittent brief ($10^3$ year-scale or shorter) gaps, such as intervals undergoing erosion or with no deposition due to repeated exposure that are hard to recognize, lead to distortion of individual astronomical cycles and increased DYNOT values. These brief gaps, if not accompanied by noise, may not be detected in the $\rho_1$ model. Long-lived ($10^5$ year scale or longer) gaps simulating unrecognized sedimentary hiatus lead to a slight increase in DYNOT and decrease in $\rho_1$; the DYNOT and $\rho_1$ models cannot be used to identify hidden gaps (Fig. 1). Long-lived recognized gaps, such as those in drill cores, lead to an increase in DYNOT, decrease in $\rho_1$, and even a discontinuity (Supplementary Fig. 1). Brief single recognized gaps are difficult to detect, hindering their recognition, but this also indicates that the sedimentary noise model tolerates single brief gaps.

**Model verification in the late Quaternary.** A sedimentary record that can provides an unambiguous verification of the sedimentary noise model has the following characteristics: contemporaneous sea-level reference data; the record is from a marginal-marine environment at a water-depth of several hundred meters (near storm-wave base); the record experienced minimal tectonic and volcanic activity with no gravity flows; high-resolution paleoclimate and sea-level proxies are accessible; and a reliable chronology has been established.

The record of global sea-level change over the past 1.4 million years[2] provides an opportunity to verify the DYNOT and $\rho_1$ models of sedimentary noise. Ocean Drilling Program (ODP) Site 1119 (Fig. 2) is located 96 km east of South Island in the Canterbury Basin, New Zealand, in 393 m water depth on the upper continental slope and 5 km seaward of the edge of the shelf[20]. The minimum water depth of the site was ~250 m during Marine Isotope Stage 2[21], which leads to an inference that paleoclimate proxy data at that site are susceptible to increased environmental noise during times of low sea level. The lithology at the site is strongly influenced by the terrigenous input from New Zealand. The drill core penetrated 514 m of silts and silty clays (glacial deposits) punctuated by muds and episodic 0.02–1.2 m thick terrigenous sands (interglacial deposits)[22]. Tectonic and volcanic activity was low in the Canterbury Basin over the past several million years[20]; the sediments are devoid of diagenesis, and evidence for gravity flows is rare[22].

At Site 1119 the gamma ray (GR) log has been interpreted as a proxy of fluctuations in clay content corresponding to variations in the rate of supply of glacial 'rock flour' from a waxing and

waning South Island ice cap[23]. The age model is based on radiocarbon dates for the past 39 ka; prior to 39 ka, 38 selected GR peaks have been correlated with tuned $\delta^{18}O$ records from ODP Sites 758 and 1143 to provide a 3.9-myr-long time scale[21,23]. The mean chronological uncertainty is less than 22 kyr over 0–1.4 Ma[24]. To reduce the dating error, fine-tuned GR log using

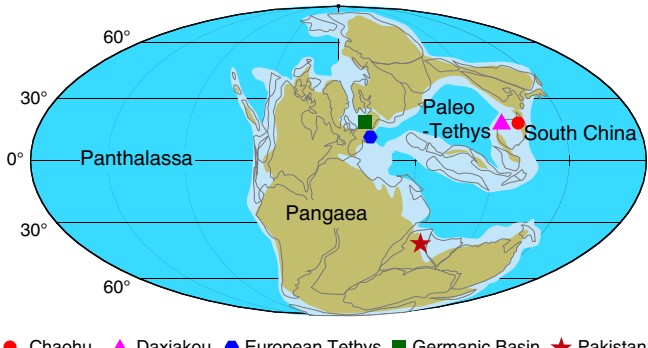

●  Chaohu  ▲  Daxiakou  ●  European Tethys  ■  Germanic Basin  ★  Pakistan

**Fig. 4** Early Triassic global paleogeographic map. The map is modified from Christopher Scotese (http://www.scotese.com) with localities of data shown in Figs. 5 and 6

astrochronology (Methods) is employed for the noise model verification.

We apply the noise model on the Site 1119 GR from 0–1.4 Ma (Fig. 3, Supplementary Figs. 2–7). DYNOT and $\rho_1$ models with a 400 kyr running window correlate well with a lowpass-filtered sea-level curve (Fig. 3). The noise model of the original GR log is also presented in the Supplementary Fig. 5. Fine-tuning the GR series leads to little change in the DYNOT and $\rho_1$ models (Supplementary Figs. 4–5), indicating that both approaches tolerate dating errors when using a relatively large 400 kyr running window. Dynamic noise in the global sea-level curve may also be linked to changes in the Earth's climate state as discussed above. DYNOT and $\rho_1$ spectra of the global sea-level curve using a 400 kyr running window show little similarity between sea-level change and dynamic noise in the sea-level curve itself (Supplementary Fig. 6), which suggests that the sources of noise at Site 1119 are different from those in global sea-level changes. For multi-million-year-long deep-time datasets from marginal marine environments, DYNOT and $\rho_1$ models can be used to provide an independent, high-resolution sea-level curve, and is demonstrated for the Early Triassic as follows.

**Model application to the Early Triassic.** Two marine sections from the South China Plate margin at Chaohu and Daxiakou (Fig. 4) provide a unique opportunity to assess a high-precision time scale conjoined with Early Triassic sea-level variations using the sedimentary noise model. The Upper Permian to Lower Triassic deep-marine successions at Chaohu and Daxiakou consist of cyclically bedded marine claystone and limestone. Sediments at both Chaohu and Daxiakou sections deposited in an offshore slope to basin setting in the early Early Triassic and the proximal ramp to outer shelf conditions in the late Early Triassic[13,25].

GR of these sedimentary rocks is affected by terrestrial clays and has been used as a proxy for continental runoff forced by climate change (Methods). The GR datasets from Chaohu and Daxiakou sections have been tuned to interpreted 405 kyr cycles[13]. A recently astronomically tuned magnetostratigraphy between South China and Germany provides an integrated time scale[13,14] to precisely correlate the reconstructed sea-level oscillations from China with European sequence stratigraphy (Fig. 5).

At Chaohu, DYNOT and $\rho_1$ show similar patterns suggesting that significantly enhanced noise occurred in the late Changhsingian, middle Induan, earliest Smithian, late Smithian, early Spathian, and latest Spathian (Fig. 6d, e). The Daxiakou section is currently 700 km distant from Chaohu (similar distance in the Early Triassic), correlates to Chaohu section (compare Fig. 6b, c

for Daxiakou and Fig. 6d, e for Chaohu). Contributing factors to the sedimentary noise model of the studied sections are provided in Supplementary Note 1 and Supplementary Figs. 11–12. This sedimentary noise modeling sets a new framework for Early Triassic sea sea levels in South China.

The sea-level variations interpreted from the sedimentary noise models at Chaohu and Daxiakou are supported by field observations at the outcrop. For example, the sequence boundaries Ol1, Ol2, and Ol3 in Fig. 6g correspond to elevated noise values at Chaohu and Daxiakou. These sequence boundaries are consistent with major lithologic changes from thin-bedded clay-rich sediments to medium-thick carbonate beds (Supplementary Fig. 8) indicating sea-level falls during clay-rich, basinal and proximal ramp conditions. Moreover, sedimentary structures indicative of high-energy and shallow water conditions, e.g., ripples and cross stratification[25], correlate with increased noise levels indicative of shallow sea levels at Chaohu (Supplementary Figs. 9-10).

Early Triassic sequence boundaries are typically presented in relative time or with ages estimated by correlation to the geologic time scale[5,7,9,26]. Here, DYNOT and $\rho_1$ modeling provides a high-resolution time frame for sea-level changes estimated directly from stratigraphy. A set of major sea-level falls during the Early Triassic have been proposed for the Boreal and Tethyan provinces in the European basins, Arctic Canada and other regions[6,9,26]. The amplitudes of these sea-level falls have been interpreted to be as much as 75 m[5,9] (although some estimates are more subdued[7]; see Supplementary Note 2 and Supplementary Fig. 13). The sedimentary noise models of sea-level change in South China correlate with these major eustatic changes (compare Fig. 6b–f), establishing the global nature and synchronicity of these $10^6$-year scale eustatic events.

**Hypothesis of aquifer eustasy.** In super-greenhouses or hot-houses with no known ice sheets, as during the Early Triassic or Late Cretaceous, ice-based models (i.e., ice sheet growth and decay) cannot explain high-amplitude sea-level variations. An alternative model that $10^5$ to $10^6$ year scale variations in continental water storage significantly changes the land–ocean water mass balance, led to the hypothesis of 'groundwater-driven eustasy', termed 'aquifer eustasy'[3,27–31] or 'limno-eustasy'[32,33]. However, this hypothesis currently has three complications: underestimation of continental water storage and its confusion with minor lake and river water volume with respect to sea-level change equivalent (<1 m)[2,30,33]; lack of direct evidence of continental water storage from the geological record; and poorly understood mechanisms and timescales of aquifer eustasy[27,28,31,32,34].

However, our understanding of the volume of continental water storage improved substantially with the 'pore space' model of Hay and Leslie[30]. According to this model, pore volume in Triassic terrestrial systems, if filled to capacity with meteoric water, could lower sea-level by more than 100 m after isostatic adjustment[30]. During the Late Cretaceous terrestrial pore space was equivalent to a global sea-level change of 200 m[30]. Compilations suggest that the present-day volume of groundwater storage is equivalent to a sea-level differential of ~320 m[35] to 330 ± 41 m[36]. Even if only a proportion of a corresponding water volume contributes to sea-level change, this change is significant[28,29,32,33].

Based on the hypothesis of aquifer eustasy, there should be a positive correlation between filled continental aquifers (high groundwater tables) and relatively high lake levels[32,33]. Consequently, non-marine sequences in the Cretaceous and Late Triassic have been proposed as a proxy for lake levels and

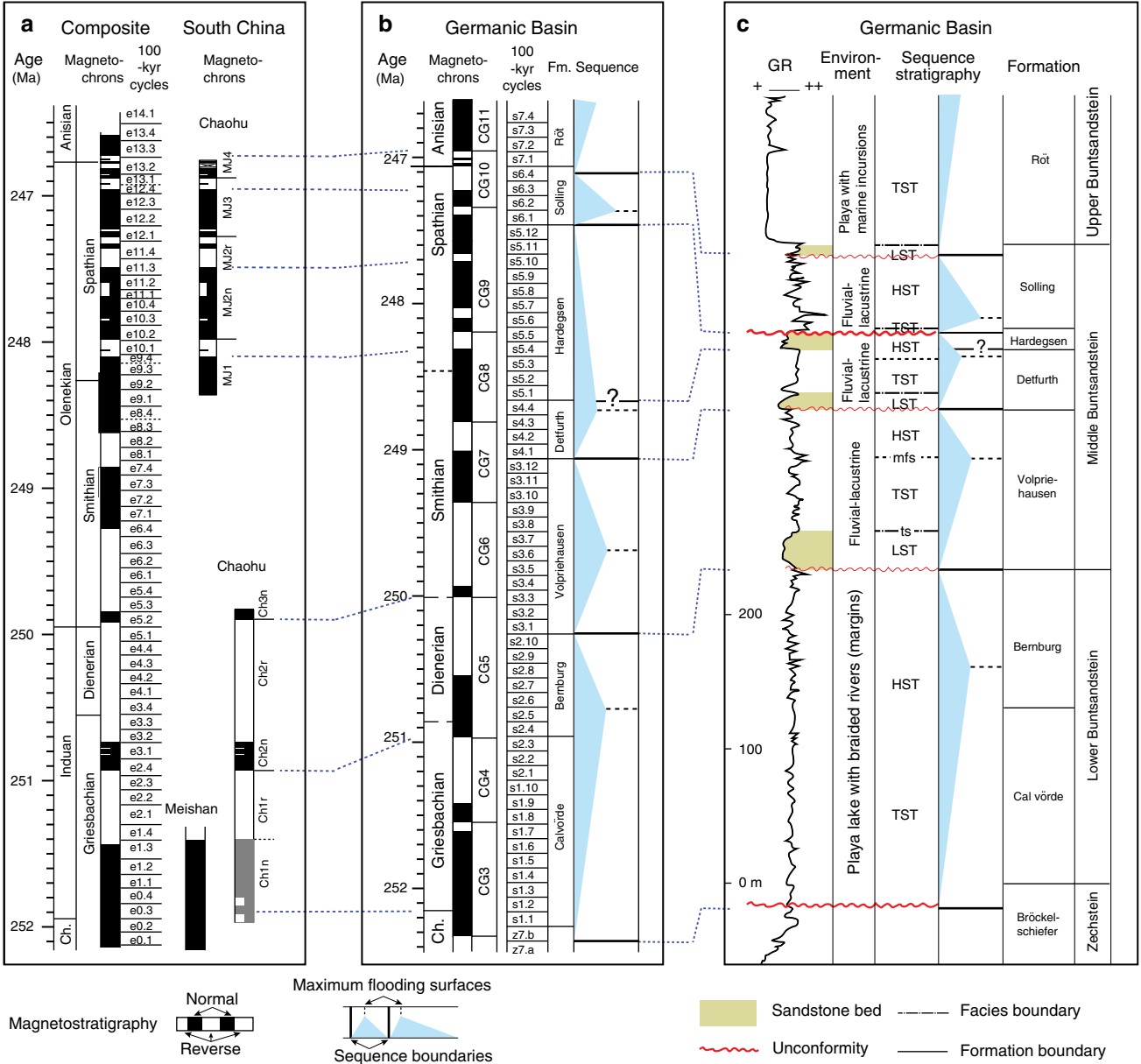

**Fig. 5** Sequence stratigraphy of the Germanic Basin. **a** 405 kyr cycle-calibrated magnetostratigraphy in South China[13]. **b** 100 kyr cycle calibrated magnetostratigraphy of the Germanic Basin[37,69,70] shown with calibrated Germanic sequences in **c**. **c** Sequence stratigraphy of the Germanic Basin[40,41] plotted in the stratigraphic domain. Dashed blue lines indicate formation boundaries in the Germanic Basin. LST: lowstand systems tract. TST: transgressive systems tract. HST: highstand systems tract. ts transgressive surface. Mfs: maximum flooding surface. Fm.: formation

continental aquifers, which moreover are out-of-phase with sea-level variations[32,33]. However, correlations of terrestrial and marine sequences in the Cretaceous and Late Triassic that support the aquifer eustasy hypothesis[32], while enlightening, suffer from a lack of reliable chronology. In this respect, the detailed astrochronology and magnetostratigraphy of the Early Triassic in South China and Germanic Basin[13,37] provide a robust time framework for correlating non-marine sequences and global sea levels (Fig. 5).

**Evidence for Triassic land–ocean water balance dynamics**. The Germanic Basin is a restricted basin with a center in northern Germany during the Early Triassic[38–40]. Groundwater tables and thus water storage variations can be inferred from sequences deposited in the lacustrine and fluvial environments of the basin.

A regressive tendency of the uppermost marginal-marine Zechstein (latest Permian) continued into the Lower Buntsandstein that was deposited in a playa-lake setting[40]. A shale-rich interval in the middle of the Bernburg Formation represents a maximum flooding surface[41]. A regional unconformity at the base of the Volpriehausen Formation has been suggested as a sequence boundary[37,40,41]. Fining-upward interbedded sandstones and shales and coarsening-upward strata of the Volpriehausen Formation were deposited in fluvial and lacustrine environments[40,41]. The overlying Detfurth and Hardegsen formations were also deposited in fluvial to lacustrine settings[41]. The widely distributed 'Hardegsen Sandstone' at the base of the Hardegsen Formation is marked by an abrupt change in lithology[37,40], which indicates a possible sequence boundary. The succeeding Solling Formation is characterized by coarse-grained fluvial sandstones, lacustrine shales, and fluvial sandstones[40]. The overlying Anisian

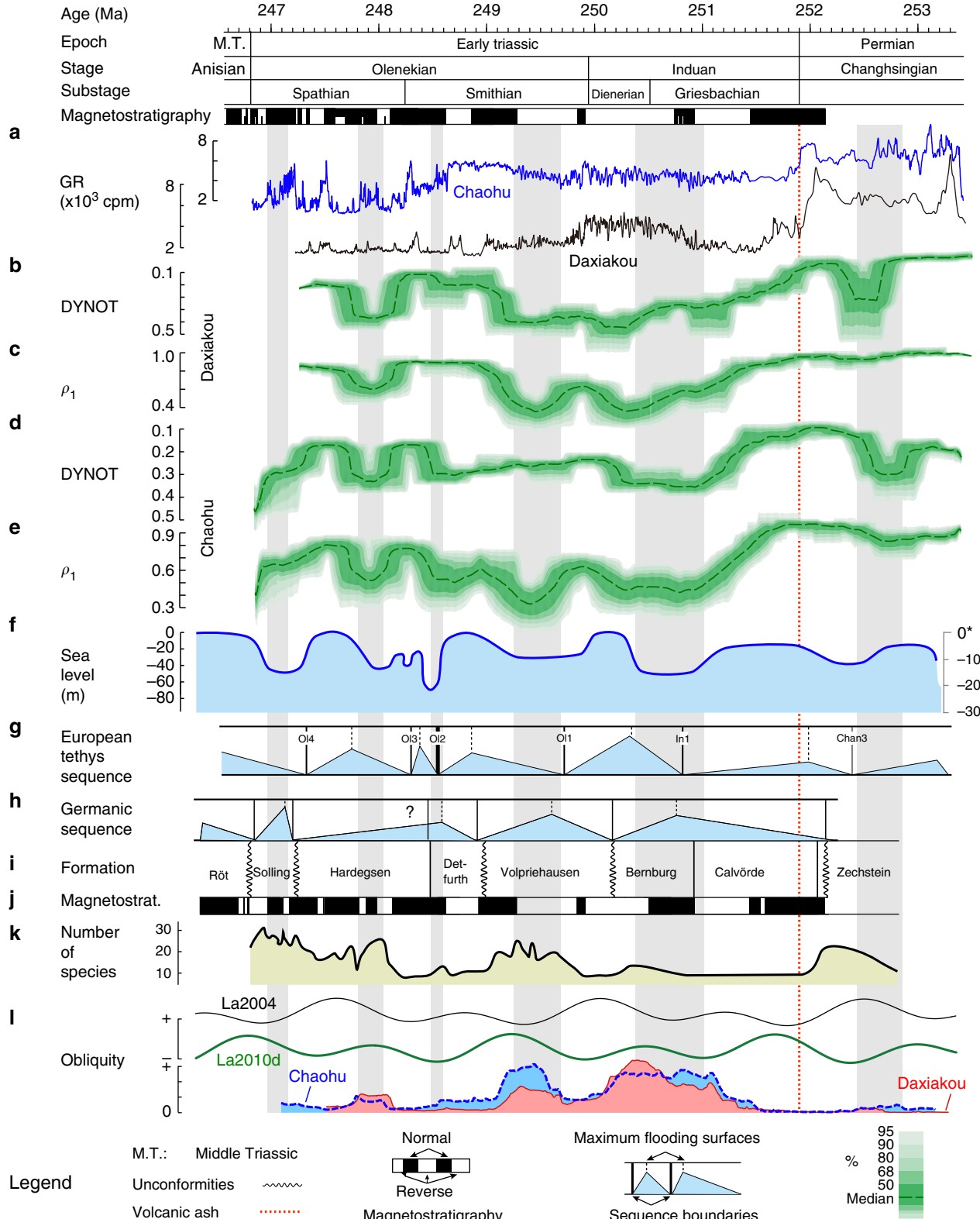

**Fig. 6** Global sea level and continental groundwater dynamics during the Early Triassic. Time scale and magnetostratigraphy are from ref.[13,14]. **a** GR series at Daxiakou (black) and Chaohu (blue) are from ref.[13]. cpm: counts per minute. **b**, **c** DYNOT and $\rho_1$ models of the GR series at Daxiakou. **d**, **e** DYNOT and $\rho_1$ models of the GR series at Chaohu. The DYNOT and $\rho_1$ models were estimated using a running window of 400 kyr (see Supplementary Tab. 1 for sample rates for each spectrum). Confidence levels were estimated by a Monte Carlo analysis with 5000 iterations. **f**, **g** Sequence and sea-level variations of the European Tethys[7,9] (*Supplementary Note 2 and Supplementary Fig. 13 for details). **h–j** Lithology, magnetostratigraphy, sequences in the continental Germanic Basin[13,40] (Fig. 5 for details). **k** Spores and pollen diversity in Pakistan[47]. **l** Earth's obliquity forcing intensity of La2004[19], La2010d[49] and in South China[31]

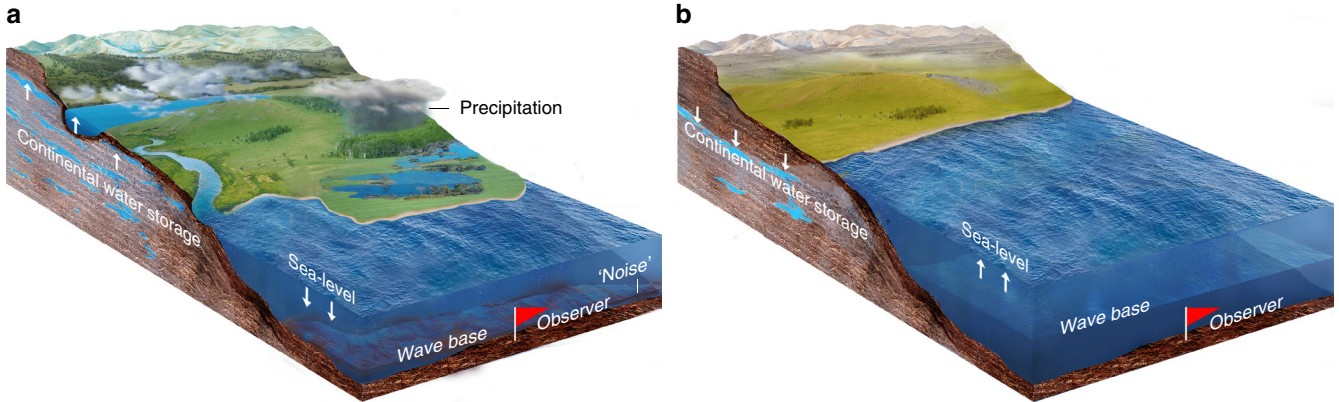

**Fig. 7** Water exchange between continental water storage and the ocean forced by astronomically forced climate change leads to major sea-level variations. **a** More moisture transferred to the continent leads to recharging of groundwater and lakes, and a flourishing terrestrial ecosystem. This results in a lowering of sea-level and wave base and more environmental noise (e.g., increased sediment mixing) at the location of observer (red flag). **b** Less moisture transferred to the continent leads to depleted groundwater and lakes, and a rise in sea-level and wave base with less environmental noise at the location of the observer. Illustrations © Hewei Duan

Röt Formation consists of fluvial sandstones and shales inter-bedded with halite deposits, deposited in a playa-like environment with multiple marine incursions[40,41].

Stratigraphic unconformities indicative of sequence boundaries show that lake levels and groundwater tables decline in the uppermost Zechstein Group, the base of the Volpriehausen, Detfurth, Hardegsen (?), Solling and Röt formations (Fig. 5). Maximum flooding surfaces are recognized in the lower Bernburg, the upper Volpriehausen, top of the Detfurth, the lower part of the Solling and the middle-upper of the Röt formations[40,41]. Orbitally tuned magnetostratigraphic correlation between South China and Germany[13,37] provides an integrated time scale and reveals that major continental water-storage falls (sequence boundaries in the Germanic Basin) occurred during the latest Changhsingian, late Dienerian, middle Smithian, latest Smithian (?), late Spathian, and end-Spathian, and rapid increases (maximum flooding surfaces in the Germanic Basin) in the late Griesbachian, early Smithian, late Smithian, and late Spathian (Fig. 5). Among these, continental water-storage falls in the latest Changhsingian, late Dienerian, middle Smithian, and latest Smithian occurred in times of quick marine ingressions. And fast increase in continental water-storage in the late Griesbachian, early Smithian and late Smithian occurred in times of sea-level falls. In other words, the ages and patterns of sequence boundaries and maximum flooding surfaces of global eustasy and Germanic terrestrial stratigraphy indicate that water masses 'see-sawed' between continental reservoirs and ocean throughout the Early Triassic hothouse (Figs. 6 and 7).

This 'see-saw' relationship becomes indistinct in the middle Spathian through the earliest Anisian (Fig. 6g, h). The interpretation is that regional tectonics may have contributed to the late Spathian unconformity between the Hardegsen and Solling formations in the Germanic Basin[40,41]. The sequences of the Anisian Röt Formation reflects frequent marine incursions[37,40], which is expected to be in-phase with global sea-level change.

Marine transgressions and regressions were also considered to drive Early Triassic sequences in the Germanic Basin[40]. However, the Germanic Basin is a land-locked terrestrial basin without connection to the open sea during the Early Triassic; marine incursions recurred only in the Anisian[38,40]. This evidence of aquifer eustasy does not require a connection via straits between the Germanic Basin and the open ocean; rather, astronomically forced million-year scale land–ocean water exchange represents an appealing interpretation.

**Astronomical forcing of land–ocean water balance dynamics.** The mechanism for high-amplitude glacio-eustasy has been linked to astronomically forced ice sheet growth and decay[2,42–44]. Paleoclimate studies indicate that the primary beat of the late Cenozoic ice sheets dynamics and high-amplitude sea-level variations is in the obliquity band[44]. In ice-free worlds, eccentricity-precession signals dominate climate change[44,45]. The evidence extends to the long-period modulations of the obliquity and eccentricity: 1.2 myr obliquity variation nodes (modulation minima) are associated with glaciation and third-order gla-cioeustatic sequences[42,43], while the 2.4 myr eccentricity modulations are associated with climate and sea-level oscillations in greenhouses[45].

The correlated magnetostratigraphy and cyclostratigraphy of the Early Triassic Germanic Basin and South China provides robust geological evidence that continental aquifer had a major impact on global sea level. Time-series analysis of the GR series and field observations in South China reveals 1.2 myr obliquity modulation cycles that are linked to Early Triassic sea levels and biodiversities[31]. Lower water storage in the Germanic Basin correlates with high sea level and decreased obliquity forcing in both South China and the La2010d astronomical solution (Fig. 6). Climate simulations and sedimentological evidence for alluvial plain and playa-lake deposits suggest that groundwater fluctuations in the Germanic Basin result from precipitation changes on the remnants of the Hercynian (Variscan)–Appalachian Mountains[38,39], i.e., a variable precipitation intensity drove changes in aquifer capacity in the Germanic Basin. This suggests the missing link between long-term obliquity forcing and million-year scale sea-level change: climate studies of the late Cenozoic icehouse have proposed that obliquity forcing dominated poleward flux of heat, moisture, and precipitation through the control of the meridional insolation gradient[46]. This mechanism may be extended to deep time with no ice sheets and to the long-period obliquity modulations. The 1.2 myr obliquity nodes are associated with reduced transportation of heat, moisture, and precipitation; the 1.2 myr obliquity variation maxima are linked to re-invigorated heat and moisture transportation and intensified precipitation[27–29,31] (Supplementary Note 3).

The time to significantly affect a groundwater reservoir after changes in global hydrologic cycle is estimated to be in the order of $10^4$–$10^5$ years[30]; this is sufficient to force 100 kyr to million-year scale variations in sea-level[32,33]. The groundwater-driven eustasy is driven by dynamic balance between filling (via precipitation) and discharge (via evapotranspiration and runoff) of continental aquifers that influenced by the hydrologic cycle[28], which is ultimately driven by paleoclimate change[3,33]. Consequently, obliquity-forced million-year scale variations in precipitation may have paced the recharging and drainage of the continental reservoirs. Recharging and drainage of groundwater and lakes indicates a significant water exchange between land and ocean, which may have paced the evolution of terrestrial ecosystems, e.g., spore and pollen diversity[31,47] (Fig. 6k).

Evidence of million-year scale obliquity-forced land–ocean water balance dynamics is strongest during the Induan-middle Smithian interval. The relationship weakens during the middle Spathian-earliest Anisian, partly because of lower obliquity band variance (e.g., Fig. 6l), the late Spathian tectonics and the earliest Anisian marine incursions in the Germanic Basin (see above). Future climate modeling and documentation of groundwater variations in other Early Triassic climate zones will further clarify this hypothesis of anti-phasing between global sea level and continental water reservoir changes.

## Discussion

Sea-level change reconstruction based on marginal marine depositional sequences is a difficult task that depends on subjective sedimentological interpretation. Monte Carlo simulations of a sedimentary noise model that links the intensity of sedimentary noise to sea-level change provides an independent method for simultaneously estimating geologic time and sea-level change from astronomically forced depositional successions. Two approaches are proposed for the sedimentary noise model, i.e., dynamic noise after orbital tuning (DYNOT) and the lag-1 autocorrelation coefficient ($\rho_1$). DYNOT and $\rho_1$ modeling of a GR series of ODP Site 1119 over the past 1.4 myr correlates with the classic low-passed $\delta^{18}O$ sea-level curve, demonstrating the efficacy of the sedimentary noise model.

Application of the sedimentary noise model to Early Triassic marine successions of South China reveals a multi-million-year history of sea-level change. The sea-level record of South China correlates with that observed in European Tethys, and these together are anti-phased with water storage variations inferred from sequence stratigraphy in the continental Germanic Basin. This geological evidence demonstrates long-term (1–2 myr) water mass exchange between the ocean and continental reservoirs (Fig. 7). The evidence suggests that obliquity-forced continental reservoir changes had a significant impact on global sea-level variations and terrestrial ecosystems during the Early Triassic and possibly throughout geologic time.

The mystery of large, million-year scale sea-level oscillations during non-glacial times challenges current knowledge of global sea-level change. Our evidence that water masses 'see-sawed' between continental reservoirs and ocean during the Early Triassic indicates that long-term obliquity-forced land–ocean water exchange is the missing link for reconciling geological records and models for sea-level change in ancient hothouses lacking ice sheets.

Sea-level rise is one of the most serious impacts of present-day climate change. In the Intergovernmental Panel on Climate Change (IPCC) assessment report, rising global sea level has been primarily linked to two factors related to global warming: land ice melting and the thermal expansion of sea-water[48]. The importance of groundwater fluctuations may be underestimated in long-term projections of global sea-level change due to lack of data or understanding of land–ocean water balance dynamics. The present-day volume of groundwater storage is equivalent to a sea-level differential of approximately 320–330 m[35,36]. Thus, as present-day Earth continues toward both warmer climate and lower obliquity angles, changes in continental aquifers should be reassessed for their contribution to global sea-level variations in long-term future projections.

## Methods

**Dynamic noise after orbital tuning**. DYNOT is assessed from the ratio of total orbital signal variance to total variance in a climate proxy time series. Time-dependent ratios of variance in the orbital band are obtained from $2\pi$ multitaper variance (power) spectra calculated along a sliding time window using the Matlab script 'pda.m'[31].

The noise after removal of orbital variance in a given time interval is:

$$R = 1 - \frac{P_{(e)} + P_{(o)} + P_{(p)}}{\sum_{i=f_{min}}^{f_{max}} P_{(i)}} \qquad (1)$$

where $f_{min}$ and $f_{max}$ are cutoff frequencies for estimation of total variance between $f_{min} = 0.001$ per kyr, and $f_{max} = 1$ per kyr. $P_{(e)}$, $P_{(o)}$, and $P_{(p)}$ are the power of eccentricity, obliquity and precession signals as defined below:

$$P_{(e)} = \sum_{i=c1}^{c2} P_{(i)} + \sum_{i=c3}^{c4} P_{(i)} + \sum_{i=c5}^{c6} P_{(i)} \qquad (2)$$

where c1, c2, c3, c4, c5 and c6 are cutoff frequencies (in per kyr) for eccentricity cycles of (405 per kyr), (125 per kyr), and (95 per kyr).

$$P_{(o)} = \sum_{i=c7}^{c8} P_{(i)} \qquad (3)$$

where c7 and c8 are cutoff frequencies (in per kyr) for obliquity cycles, which is (40.9 per kyr) in the past 1.4 myr, and (33 per kyr) at 249 Ma[19,31,49].

$$P_{(p)} = \sum_{i=c9}^{c10} P_{(i)} + \sum_{i=c11}^{c12} P_{(i)} + \sum_{i=c13}^{c14} P_{(i)} \qquad (4)$$

where c9, c10, c11, c12, c13 and c14 are cutoff frequencies (in per kyr) for precession cycles, which are (23.6 per kyr), (22.3 per kyr), and (19.1 per kyr) over the past 1.4 myr. This constitutes the DYNOT approach to sedimentary noise modeling. Proxy series in the stratigraphic domain should be first calibrated to the time domain to reduce effects of variable sedimentation rate, which can lead to frequency splitting in the orbital band. However, a variable sedimentation rate is tolerable, because the DYNOT model adopts a relatively wide passband for assessment of Milankovitch forcing signals.

**Lag-1 autocorrelation coefficient**. The lag-1 autocorrelation coefficient ($\rho_1$) is given by ref.12:

$$\rho_1 = \frac{\sum_{i=2}^{n} x_{(i)} * x_{(i-1)}}{\sum_{i=2}^{n} x_{(i-1)}^2} \qquad (5)$$

where, $x$ is the orbitally tuned stratigraphic proxy series. The advantage offered by $\rho_1$ is that it evaluates time series directly and is independent of frequency band selections.

**Relationship between DYNOT and $\rho_1$ models**. The DYNOT model removes interpreted orbital variance from what are typically the lower frequencies in cyclostratigraphic power spectra; the proportion of non-orbital variance in the power spectrum (equation (1)) is taken to represent uncorrelated noise. In comparison, the $\rho_1$ model is a simple measure of the distribution of variance across the power spectrum. The most renowned application of $\rho_1$ is for first order autoregressive modeling of red noise spectra[11]. For lower values of $\rho_1$, variance is more uniformly distributed across the power spectrum; for higher values of $\rho_1$, variance occurs preferentially in the low frequencies (e.g., Fig. 4.18 in ref. 50). That is, $\rho_1$ measures the 'redness' of variance as a function of frequency: less red (lower $\rho_1$) values are taken to represent more uncorrelated noise; higher $\rho_1$ values are associated with 'redder' spectra, and to lower dynamic noise. It is therefore no accident that the DYNOT and $\rho_1$ models move in opposite directions to mark the presence of uncorrelated noise.

**Uncertainty analysis of the sedimentary noise model**. We use a Monte Carlo method to evaluate uncertainty of the DYNOT and $\rho_1$ models of stratigraphic noise. There are two uncertainties associated with the $\rho_1$ model, i.e., sampling rate

and running window size, and 16 uncertainties with the DYNOT model, e.g., sampling rate, running window size, and 14 bandpass cutoff frequencies for 7 target orbital frequencies (Supplementary Tab. 1). Median DYNOT and $\rho_1$ model values and their 50%, 68%, 80%, 90%, and 95% significance intervals for any given time are estimated by Monte Carlo simulation with 5000 to 10,000 iterations. The uncertainties and their ranges are discussed below.

Sampling rate is a complex issue for deep-time paleoclimate data with uncertain timescales, especially for data that have not been measured at a uniform sample spacing, or data that have been time-calibrated[50]. This is the case for all data in this study (Supplementary Tab. 2).

For the DYNOT model, non-uniformly sampled paleoclimate time series can be interpolated to a uniform sampling rate to allow application of powerful time series methods for uniformly sampled time series, e.g., the multitaper (MTM) power spectrum[51]. Here, a Monte Carlo method of hypothesis testing using the MTM power spectral analysis is undertaken, and so resampling must be applied.

Sampling rates of proxy datasets in time are always greater than zero and so are non-normally distributed. Therefore we selected the Weibull distribution[52] to represent sampling rate distributions for the uncertainty analysis of the DYNOT model. The Weibull probability plot[52] of sampling rates of the gamma ray time series at Site 1119 is nearly linear, indicating that the sampling rates are reasonably fit by a Weibull distribution (Supplementary Fig. 2). To avoid ultralow or ultrahigh, unrealistic sampling rates we set the 5th and 95th percentiles of sampling rates as lower and upper limits of Monte Carlo-generated Weibull-distributed sampling rates.

Based on definition of the $\rho_1$ model (equation 5), upsampling to increase sampling rate leads to artificially high $\rho_1$ values. In comparison, downsampling to decrease sampling rate results in relatively low $\rho_1$ values. To address this problem, we apply the uniform distribution to represent sampling rates for the uncertainty analysis of the $\rho_1$ model. To avoid ultralow and ultrahigh, inappropriate sampling rates we set the 95th percentiles of sampling rates ($sr_1$) as the lower limit of Monte Carlo-generated uniformly distributed sampling rates and 1.5–2.0 times $sr_1$ as the upper limit (Supplementary Tab. 2).

The dynamic stratigraphic noise spectrum is calculated with a running time window across a uniformly sampled climate proxy series. Different windows can affect DYNOT and $\rho_1$ results in two ways:

First, a large window will shorten the number of calculated model values, and a small window will generate more calculated model values, $N_r = N_{data} - N_{win} + 1$, where $Nr$ is total number of model values for a given simulation, $N_{data}$ is total number of interpolated data points, and $N_{win}$ is number of points in the running window. Thus, smaller $N_r$ compared to $N_{data}$ leads to a 'no data' effect at the beginning and end of the noise output. To avoid this problem, the dynamic noise model randomly shifts and plots simulation results of a single iteration at the same time scale of the dataset, although this generates relatively smoothed dynamic noise spectra when a gap is shorter than $2 \times N_{win}$ (e.g., the gap $\eta$ in Supplementary Fig. 1).

Secondly, modeling with a small running window generates higher frequency results (Supplementary Fig. 7), however, the variance of low-frequency cycles and total variance diminish simultaneously, which leads to increased uncertainty. A small running window also increases the MTM power spectrum bandwidth (i.e., reduces frequency resolution) in the DYNOT model.

The expected sea-level variations of interest in the Early Triassic are $10^4$ to $10^6$ year scale, i.e., the fifth to third-order scale[2,5,45], therefore a comparable or shorter time window (e.g., 300–500 kyr, or shorter) should be adopted for the modeling. A running window of 400 kyr and randomized windows within a 300–500 kyr range show small differences (Supplementary Figs. 1 and 3), indicating that both window settings are appropriate. A small running window of 100 kyr for the ODP Site 1119 generates higher frequency results in the past 1.4 myr (Supplementary Fig. 7), although shortcomings is presented above.

Uncertainties derived from different filter cutoff frequencies apply to the DYNOT model only. For the definition of the noise for orbital tuning ($R$) in equations (1–4), cutoff frequencies and bandwidths are crucial for variance estimation of eccentricity, obliquity and precession signals. In some circumstances, such as absence of sediment at maximum flooding surfaces and/or short-lived exposure related to brief sea-level fall, the paleoclimate proxy might be known well enough for hiatuses to be detectable[53]. Multiple hiatuses at random spacing can lead to broadened and shifted frequencies[53]. The effects of a variable sedimentation rate can also lead to frequency splitting in the astronomical bands. Therefore, definition of cutoff frequencies can introduce uncertainties to the DYNOT model.

Target astronomical frequencies can be estimated from the power spectrum of an astronomical solution, e.g., La2004[19] for given time interval (Fig. 1). During the past several million years, astronomical cycles are dominated by 405 kyr, 125 kyr, and 95 kyr eccentricity cycles, 40.9 kyr obliquity cycles, and 23.6 kyr, 22.3 kyr, and 19.1 kyr precession cycles[19,49]. During the Early Triassic, Milankovitch cycles were dominated by 405 kyr, 125 kyr, and 95 kyr eccentricity cycles, 33 kyr obliquity cycles, and 21 kyr, 20 kyr, and 17 kyr precession cycles[19,31,49].

In equations (1–4), c1 to c14 are cutoff frequencies for 3 eccentricity cycles, 1 obliquity cycle, and 3 precession index cycles. One could set cutoff frequency ranges to a minimum of ±20% of the target frequencies. For example, the obliquity cycles for the past 1.4 myr have a frequency of $0.0244 \pm 0.0049$ per kyr. However, the MTM power spectrum bandwidth resolution of a single computation in the DYNOT model can be much wider than the above set frequency ranges if the

running window is relatively short. We vary each cutoff frequency assuming a uniform distribution with cutoff frequency ranges at ±90% to ±120% bandwidth (Supplementary Tab. 1). Here the bandwidth (bw) equals $nw/N_{win}$, where nw is time-bandwidth product of discrete prolate spheroidal sequences used in the multi-tapers, and $N_{win}$ is length (in data points) of the running window.

**Stratigraphy of the Chaohu section.** The Chaohu section near Chaohu City, Anhui Province exposes an Upper Devonian to Middle Triassic sedimentary succession. Chaohu is located on the north margin of the South China plate[54]. Depositional environments range from deep basin to base of slope/lower slope facies during the latest Permian to Early Triassic[54]. During the latest Permian to early-middle Early Triassic Epoch, the Chaohu area was in a deep basinal environment with deep-water ammonoid and conodont species[54,55]. The Changhsingian (latest Permian) Dalong Formation is composed of grayish-black cherty beds and cherty mudstone with deep-water assemblages including the bivalve *Hunanopecten* sp., ammonoid *Pseudotirolites* sp. and radiolarian *Flustrella* sp.[56]. The Lower Triassic Yinkeng, Helongshan and Nanlinghu formations comprise cyclic beds of marine mudstone (or shale) and marlstone. The clay content decreases and carbonate content increases significantly up section, supporting the hypothesis that the Lower Yangtze sedimentary province was shallowing during the late Early Triassic[54,57] due to the collision of the North China and South China plates in the Middle-Late Triassic[58].

Sedimentology of the Chaohu section indicates significant sea-level changes throughout the Early Triassic. Li et al.[57] interpreted a maximum flooding surface in the upper of the Induan Stage at Chaohu. Increased medium-bedded limestone and hummocky cross-stratification near the Induan-Olenekian boundary (Supplementary Figs. 8–10) are interpreted as indicating a relative sea-level fall in the middle Yinkeng Formation. The 20 m thick Helongshan Formation consists of limestone interbedded with green shale and micrite limestone[54]. Thick-bedded limestone with hummocky cross-stratification at the base of the Helongshan Formation indicates a relatively high-energy depositional settings[57]. The abrupt change in lithology from mud-rich sediments at the top of the Yinkeng Formation to thick-bedded limestone in the overlying Helongshan Formation is interpreted as a response to a drop in sea-level. The top of the Helongshan Formation is dominated by dark gray medium-bedded micritic limestone interbedded with black shale and calcareous shale. These characteristics, together with a prominent gamma ray maximum suggest that the top of the Helongshan Formation represents a maximum flooding surface.

The overlying Spathian Nanlinghu Formation is comprised of relatively thick-bedded carbonates. The base of the Nanlinghu Formation has cross-bedding indicative of current ripples, wavy cross bedding and abundant trace fossils, such as *Planolites*, *Palaeophycus*, *Arenicolites*, *Diplocraterion*, *Chondrites*, *Thalassinoides*, and *Monocraterion*[25]. This evidence, together with absence of pyrite (Supplementary Figs. 9-10), suggests an oxidized[59], high-energy depositional environment during the sea-level drop in the earliest Spathian. Li et al.[57] also interpreted a maximum flooding surface in the lower-middle Nanlinghu Formation, which is characterized by thin-bedded limestone, horizontal stratification and high gamma ray values (Supplementary Figs. 9–10).

The Nanlinghu Formation is overlain by the lower Anisian Dong Ma'anshan Formation of a basal brecciated (karstic?) limestone, indicating a relative sea-level fall in the late Spathian (see also ref. 57). This sea-level fall at the northern margin of the South China platform appears to be confirmed by the coeval Guandao section in the Nanpanjiang Basin of South China[14]. The late Spathian at Guandao is also characterized by presence of breccia and thick limestone and relatively low gamma ray responses[13,14], likely associated with sea-level fall. The Dong Ma'anshan Formation is succeeded by Middle Triassic evaporites and non-marine terrestrial deposits and Late Triassic fluvial-lacustrine deposits in the Lower Yangtze region[60].

**Stratigraphy of the Daxiakou section.** The Daxiakou section is located at 6 km east of Xiakou town of Xingshan County, Hubei Province. The lithology at the Daxiakou section is similar to that of the Chaohu section, but with thinner limestone beds and fewer mudstone beds[13]. The latest Permian Dalong Formation is composed of black shale and mudstone interbedded with multiple volcanic ash layers. The Induan-Olenekian Daye Formation is dominated by grayish thin-bedded limestone rhythmically interbedded with marls and mudstone in the lower part, and with thin-bedded limestone in the middle and upper parts[13,61]. The overlying Middle Triassic Jialingjiang Formation consists of gray limestone, taupe argillaceous dolomite and dark gray dolomitic limestone[62].

**Gamma-ray as paleoclimate proxy.** Gamma ray (GR) in sedimentary rocks is a proxy for terrestrial input into the marine depositional environments of our studied sections[13,31]. GR of sediments is dominated by potassium (K), uranium (U) and thorium (Th)[63]. K is common in many minerals such as clays, feldspar, mica, and chloride salts. U and Th are concentrated in a number of sedimentary host minerals including clays, feldspar, heavy minerals, and phosphate, and U is often concentrated in organic matter[63]. At ODP Site 1119 the GR has been interpreted as a proxy of fluctuations in clay content corresponding to variations in the rate of supply of glacial 'rock flour' from a waxing and waning South Island ice cap[23]. In the Early Triassic, high GR values of post-extinction interval sedimentary rocks are

attributed to clay-rich sediments, while low GR values are linked with coarser-grained rocks and carbonates. Variable clay content can be related to climate change from Milankovitch forcing, e.g., during high eccentricity hotter summers relative to winters may have resulted in intensified weathering and stronger monsoonal climate. More frequent rainfall and runoff would result in greater clay influx into the marine depositional environment, high GR and U, and vice versa[13,31]. Deep weathering of outcrops can result in leaching of K and U[64], however, due to the use in the present study of new road-cut sections at Daxiakou and Chaohu dissolution from weathering is minimized.

**Astrochronology of gamma ray series at ODP Site 1119**. To decipher the impact of dating error to the sedimentary noise model, we applied two age models at ODP Site 1119 in this study: the original chronology from refs. [21,23] and astro-chronology based on fine-tuning to monotonic 40.9 kyr obliquity cycles. The original age model at Site 1119 is based on radiocarbon dates for the past 39 ka; prior to 39 ka, 38 selected gamma ray peaks were correlated with tuned $\delta^{18}O$ records from ODP Sites 758 and 1143 to provide a 3.9-myr-long time scale[21,23]. The mean chronological uncertainty is less than 22 kyr over 0–1.4 Ma[24]. The chronology at ODP Site 1119 has elevated uncertainties (up to 100 kyr) at ca. 0.25, 0.38, 0.57–0.78, and 1.19–1.45 Ma[24]. We thus elected to fine-tune the original gamma-ray series in refs.[21,23] using filtered obliquity cycles.

The 40.9 kyr obliquity cycles are predominant from 900 ka to 1400 ka in the gamma-ray variations at ODP Site 1119; they also occur in the gamma-ray series from 0 to 900 ka (Supplementary Fig. 4). Therefore, we filtered the obliquity cycles from the gamma-ray time series in refs. [21,23] and constructed an age model based on the filtered 40.9 kyr obliquity cycles (Supplementary Tab. 3). This age model was used for the fine-tuning of the original gamma-ray series. We then applied the sedimentary noise model to the fine-tuned gamma-ray series (Fig. 3 and Supplementary Figs. 3, 5 and 7).

**Astrochronology methods**. The identification of the obliquity signal in the gamma ray logs proceeded as follows: The gamma-ray series were pre-whitened using Matlab script 'smooth.m' to estimate and subtract an 800-kyr long-term LOWESS curve[65]. Evolutionary fast Fourier transform (FFT) spectrograms for inspecting stratigraphic frequencies and patterns of the untuned and tuned series were computed using 'evofft.m'[50]. The gamma ray logs were analyzed with the multitaper method (MTM) spectral estimator[51] using Matlab's 'pmtm.m'. Conventional red noise models of the time series were estimated using the Matlab script 'redconf.m'[66]. Based on the inferred wavelengths of prominent cycles, Gaussian bandpass filtering was applied in Matlab to isolate potential orbital parameters using 'gaussfilter.m'[50]. The original series was fine-tuned using 'depthtotime.m' Matlab script[50] based on the 40.9 kyr obliquity cycles identified by filtering.

**Astronomical solutions**. La2004[19] and La2010[49] are astronomical solutions of Earth's eccentricity, obliquity, and precession index for the 0–250 Ma. Solutions La2010a, b and c are based on the INPOP08 ephemeris while La2010d is based on INPOP06[49]. INPOP06 was later found to be more precise than INPOP08[67], (INPOP = Intégration Numérique Planétaire de l'Observatoire de Paris). While a strictly accurate astronomical solution is not available for times before 50–60 Ma[49], the 1.2 myr obliquity modulation persists in both La2004 and La2010d solutions through 249 Ma[31]. La2010d obliquity modulations were obtained using the procedure in refs.[31,68].

**Code availability**. The software (*Acycle v0.1.3*) that supports the findings of this study is available from the corresponding author M.L. on request.

**Data availability**. The gamma ray data at Chaohu, Daxiakou sections and ODP Site 1119 can be found at https://doi.org/10.1016/j.epsl.2016.02.017 and https://doi.org/10.1126/science.1093726.

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

## Acknowledgements

We thank Yang Zhang, Zuoyan Zou, Zhong-Qiang Chen for field assistances. We thank the referees for their detailed comments and instructive suggestions. The Triassic sea-level change and uncertainty analyses benefitted from discussions with Bilal Haq and Zhu Liu. This study was supported by the National Natural Science Foundation of China (No. 41772029, 41322013), Natural Science Foundation for Distinguished Young Scholars of Hubei Province of China (2016CFA051), and 111 Projects (B14031 and B08030). M.L. acknowledges support from the China Scholarship Council (201406410029) for Ph. D. work at Johns Hopkins University.

## Author contributions

M.L. and C.H. designed research. M.L. developed the DYNOT and $\rho_1$ models with contributions from L.H. M.L. analyzed data. The paper was written by M.L. and L.H. with contributions from C.H. and J.O.

## Additional information

**Competing interests:** The authors declare no competing interests.

