## [Peer review file · Nature Communications]

Reviewers' comments:

Reviewer #1 (Remarks to the Author):

Dear authors

I like the paper for its rigour and its theme.

The major claim of the paper is that, during hot-house conditions, water storage on the continents and water storage in ocean basins is anti-phased. This correlation is used to suggest that continental water storage-and-release is responsible for high amplitude, obliquity-driven sea level changes during the Triassic. To get to this point the authors use a 'dynamic sedimentary noise model', which appears to be a novel methodological contribution in its own right. However I must say that I am in no position to evaluate the robustness of the methodology and techniques, but I can say that the analysis appears to be very detailed.

The data support the idea that continental water storage could have a major influence on global sea level change. This appears to be new but the idea is not a completely new one. No doubt these results are of significant interest to many branches of Earth, atmospheric and hydrological sciences, but the broad relevance of the work is not highlighted.

The conclusions are poorly written and fail to focus on the real outcome of the work. This may be because there is a dual focus to the paper: (1) Methodological, and (2) The water storage and release story. Overall the paper sways towards the methodological aspect, which dilutes the water storage aspect (the intended focus of the paper). The paper seems to skip key explanations of what data was collected and how, e.g. the Germanic fluvial and lacustrine dataset. Clearly this dataset is key but I was left wondering "what was actually measured to allow the authors to reconstruct water storage over time?". Yet the authors went into a lot of poorly-explained detail about how their worked'. Im not necessarily saying that this material should be removed, but rather to focus on the broad scientific issue of 'water storage' and refer the authors to details of the model in the SI. This would free up some space for you to 'beef-up' your results section (evidence) and discussion of the origin continental water storage versus water storage in ocean basins is anti-phased. I realise that you MUST demonstrate the robustness of your model to unequivocally demonstrate the 'antiphase', so it is a difficult one. I would like more clarity of the methods and results of the continental Germanic data collection.

The work is 80% convincing, which could be increased to 95% if my comments are answered. The authors do a rigorous job in their data analysis on the marine sections, but to demonstrate the point of the paper (i.e. sea-saw water storage fluctuations) more clarification is need on what data and analysis was done on the Germanic terrestrial/lacustrine section. This was unclear to me. Furthermore, I was not satisfied with the mechanism the authors put forward for continental storage and release over the timescales assumed. What are the mechanics of this process other than 'obliquity forcing'. Can O-isotopes of the terrestrial successions offer a future dataset that can link terrestrial to marine? How long does it take to fill and release water to and from a continental reservoir? Could this timescale be sufficient to explain any anti-phased relationship or amplification? How are these reservoirs filled and emptied?

The paper will certainly influence thinking in the field but requires more careful crafting of arguments and a clearer explanation of: 1) data collection and analysis at Germanic site; 2) the linkage between the Germanic site and marine sections; and 3) the mechanism (and timescales) responsible for storage and release in continental reservoirs.

Rob Duller

Reviewer #2 (Remarks to the Author):

This manuscript presents a novel model approach to robustly correlate sedimentary sequences governed by sea-level fluctuations. It aims at extracting cyclic sea-level signals and separating them from a variety of noise which previously prevented reliable correlation in sequence stratigraphy. The study benefits from a convincing test using a young record of well-established sea-level changes.

The paper then goes a significant step further to what I consider its most important outcome: It shows on the level of model-correlation that in the early Triassic - a prime study interval for non-glacial sea-level fluctuations - global ocean levels changed in anti-phase with continental lake levels. This supports the hypothesis of aquifer eustasy by suggesting, for the example of the Germanic Basin, that continental precipitation was the main factor responsible for rises and falls of this playa lake.

This paper will be of broad interest to stratigraphers, sedimentologists, paleo-climatologists etc. Importantly, it is very much needed for correct correlation attempts in sequence stratigraphy, and by providing evidence of the aquifer-eustatic process the present paper will influence thinking in this broad field of geoscience.

The manuscript is well-organized, technically well-performed, and I consider it mostly publishable at its present form.

The statistical approach of the presented new DYNOT model is robust and has been exposed to a strict Monte Carlo uncertainty analysis.

I recommend the following additional short discussion on the forcing of the Germanic Basin lake-level: The authors should address the previously postulated connecting straits between the Germanic Basin and the Tethys ocean, like the Schlesian Strait. As it is shown that global ocean-level rise was in anti-phase with the lake-level it follows, that marine ingressions through these straits cannot have been a significant contribution of water to the lake, as has been previously thought.

This paper stimulates exciting new research questions along the lines of the above thought. Obviously it should not lose conciseness by delving much into these aspects, yet I recommend placing a note on the need for research addressing these questions.

Using the attached annotated ms-file, please correct minor errors in writing and respond to the minor comments. Regarding data availability, it would be preferable if the data of this study could be stored in an online database.

Jens Wendler

Reviewer #3 (Remarks to the Author):

Li et al provide a novel method for estimating medium term (1 to 2 Myr) variations in sea level using analysis of time series that are proxies for sediment composition. As explained below I would support publication after minor revision.

Firstly they developed a new method for analysis demonstrated for a Pleistocene to Recent section from ODP Leg 181 off New Zealand and then applied the method to two Chinese natural gamma

ray logs from the Triassic. The method simply involves removing the quasi-regular orbitally-driven components of the time series and then find the time-varying ratio of the "non-orbital" variance to the total (untreated) time series variance. As a check they have used the time-varying lag-1 autocorrelation (which is largely independent of the regular components). Their method is novel and, accepting their interpretation that the results indicate 1-2 Myr variations in sea level, potentially of great value.

It should be pointed out that they have only used one Pleistocene case of just 1.4 Myr long to test their data. Much longer sea-level related records from passive continental margins could be used for testing (e.g. the New Jersey margin investigated by the IODP Leg 313 following up earlier ODP studies). On the other hand, this idea deserves investigation so I'm happy to go along with their argument that they have recovered a long-term sea level change off New Zealand in the Pleistocene and that they have correctly interpreted the meaning of the Chinese results too.

Secondly having judged their Chinese results to represent global sea level variations they explain the variations themselves in terms of 1-2 Myr cycles in palaeoclimate as driven by long terms of the orbital obliquity.

Overall the treatment is detailed and convincing, but I did not immediately understand the key points listed above because there are some aspects of the paper that could do with clarifying. The suggestions made here are provided to help improve clarity. They do NOT represent a criticism of the arguments or of the conclusions.

Suggestions for improvement:

1) The title could be much clearer. The author describe the "non-orbital" variance extracted with their main method as "Dynamic sedimentary noise". This is a strange use of the word dynamic and as it is not standard I think most readers will not have a clue what the paper refers to from the title. Similarly the second part of the title refers to "land-ocean water balance dynamics". This also fails to connect the reader with the idea that 1-2 Myr sea level variation in the Triassic was driven by long-term obliquity variations. Why not change the title to something like: "Sedimentary noise and Triassic sea levels linked to long-term obliquity forcing"?

2) The method is said to remove the orbitally forced components of the compositional variability from the time series (built into the term DYNOT). However, strictly it only removes the orbital components between 405 kyr and 18 kyr. The reasoning that Triassic (hothouse) sea level variations over 1-2 Myr are related to long-term obliquity forcing of climate means the authors think there are long-term orbital components in the DYNOT estimates. Therefore it needs to be made much clearer that the DYNOT estimates only eliminate the "short term" orbital components. To clarify this the authors should change the abstract on line 1 from "high-frequency sea-level" to "high-frequency (1/1-2 Myr) sea-level" [though I personally don't think of 1/1-2 Myr as high frequency]. Similarly on line 32 of the abstract change "astronomically forced water mass exchange" to "long-period (1-2 Myr) astronomically forced water mass exchange".

Other points:

The authors have been careful to list caveats from line 106 page 5. However, they have not mentioned that interpolation of irregularly spaced data (inc. natural gamma ray counts at ODP1119 put onto a time scale) leads to increased lag-1 autocorrelation/decreased DYNOT (avoidable by processing the uninterpolated data using the Lomb-Scargle transform).

Page 6 line 132 "may not detected" > "may not be detected".

Page 8 line 162 "Supplementary Fig. 5" > "S... Fig. 4)"

Page 8 line 166 Ditto Fig. 6" > Ditto Fig. 5)"

Page 12 line 251-252 "between obliquity forcing and sea-level change" > "between long-term obliquity forcing and million-year sea-level change".

Page 12 line 262 "Evidence of obliquity-forced" > Evidence of million-year obliquity-forced"

Page 13 line 271 "lower obliquity" > "lower average obliquity"

Page 13 line 281 "the classic d18O" > "the classic low-passed d18O".

Page 13 line 287 "demonstrates dynamic water mass exchange" > "demonstrates long-term (>1 Myr) water mass exchange" [how can water mass exchange NOT be dynamic??]

Page 14 line 308 End of sentence: add ref to 19, 35, 48 (as used on p18 line 401).

Page 16 lines 347-351 Again be careful interpolation leads to increase in lag-1 autocorrelation (noted above).

Page 35 line 720 "interpolated time series" > "interpolated, tuned time series".

Fig 6 Could easily be dropped altogether (the reader can easily visualize the idea of transfer of water onto and off continents affecting sea level).

graham.weedon@metoffice.gov.uk 22nd Sept 2017

Response to referees (manuscript NCOMMS-17-21491)

All line numbers are original manuscript line numbers.

Response to reviewer #1: Dr. Rob Duller

Comment:

I like the paper for its rigour and its theme. The major claim of the paper is that, during hot-house conditions, water storage on the continents and water storage in ocean basins is anti-phased. This correlation is used to suggest that continental water storage-and-release is responsible for high amplitude, obliquity-driven sea level changes during the Triassic. To get to this point the authors use a 'dynamic sedimentary noise model', which appears to be a novel methodological contribution in its own right. However I must say that I am in no position to evaluate the robustness of the methodology and techniques, but I can say that the analysis appears to be very detailed.

The data support the idea that continental water storage could have a major influence on global sea level change. This appears to be new but the idea is not a completely new one. No doubt these results are of significant interest to many branches of Earth, atmospheric and hydrological sciences, but the broad relevance of the work is not highlighted.

Reply: The original manuscript had a paragraph on long-term future projections of sea level. We have now moved that paragraph to the end of the paper. The paragraph has also been slightly extended to highlight the broad relevance of the work.

The conclusions are poorly written and fail to focus on the real outcome of the work. This may be because there is a dual focus to the paper: (1) Methodological, and (2) The water storage and release story. Overall the paper sways towards the methodological aspect, which dilutes the water storage aspect (the intended focus of the paper). I realise that you MUST demonstrate the robustness of your model to unequivocally demonstrate the 'antiphase', so it is a difficult one.

Reply: We needed to demonstrate the robustness of the model before applying it to the Early Triassic. As explained in the introduction, two leading methods (i.e., oxygen isotope and sequence stratigraphy) for sea-level reconstruction in deep time have significant problems. Therefore, this "sedimentary noise" model was developed. Description, discussion and verification of this new model were necessary to lay down a solid foundation for the reconstruction of global sea-level during the Early Triassic.

The paper seems to skip key explanations of what data was collected and how, e.g. the Germanic fluvial and lacustrine dataset. Clearly this dataset is key but I was left wondering "what was actually measured to allow the authors to reconstruct water storage over time?". Yet the authors went into a lot of poorly-explained detail about how their worked'. Im not necessarily saying that this material should be removed, but rather to focus on the broad scientific issue of 'water storage' and refer the authors to details of the model in the SI. This would free up some space for you to 'beef-up' your results section (evidence) and discussion of the origin continental water storage versus water storage in ocean basins is anti-phased.

I would like more clarity of the methods and results of the continental Germanic data collection. The work is 80% convincing, which could be increased to 95% if my comments are answered. The authors do a rigorous job in their data analysis on the marine sections, but to demonstrate the point of the paper (i.e. sea-saw water storage fluctuations) more clarification is need on what data and analysis was done on the Germanic terrestrial/lacustrine section. This was unclear to me.

Reply: In this revision, the hypothesis of aquifer eustasy is now presented in detail. Problems with the aquifer eustasy hypothesis are also discussed. We have rewritten the section about linkages among sequences, lake level, groundwater table, and continental water storage in the Germanic Basin. We have now also moved one key figure on the Germanic sequences from the SI of the original submission to the main paper (new Fig. 5). Previous studies have demonstrated that sequences in a closed terrestrial basin indicate relative changes in lake level and groundwater table; therefore sequences can be used as an proxy for ancient continental aquifer changes. Consequently, our work draws on the widely accepted sequence stratigraphy of the Germanic Basin.

Furthermore, I was not satisfied with the mechanism the authors put forward for continental storage and release over the timescales assumed. What are the mechanics of this process other than 'obliquity forcing'. Can O-isotopes of the terrestrial successions offer a future dataset that can link terrestrial to marine? How long does it take to fill and release water to and from a continental reservoir? Could this timescale be sufficient to explain any anti-phased relationship or amplification? How are these reservoirs filled and emptied?

The paper will certainly influence thinking in the field but requires more careful crafting of arguments and a clearer explanation of: 1) data collection and analysis at Germanic site; 2) the linkage between the Germanic site and marine sections; and 3) the mechanism (and timescales) responsible for storage and release in continental reservoirs.

Reply: The mechanics of the astronomically forced recharge and discharge of the continental aquifer has been rewritten in this revision. We now present more details for the evidence of obliquity forcing in the South China marine sections, precipitation in the continental Germanic basin, the timescale of the aquifer-eustasy process, and the filling and discharge dynamics that drive continental aquifer variations. As to O-isotopes and groundwater variations in terrestrial basins, Holocene-Quaternary speleothems (e.g., Hulu Cave) have been useful in this regard. Alternatively, "future climate modeling and documentation of groundwater variations in other Early Triassic climate zones will further clarify this hypothesis of anti-phasing between global sea level and continental water reservoir changes".

Comment: Line 23: "*dynamic sedimentary noise model*". Ease the reader in slowly. This is meaningless terminology; at this stage.

Reply: We have removed the term "dynamic" – also contentious with the other reviewers.

Comment: Line 52: "*Developments in sequence stratigraphy have greatly clarified the origin of genetically related sedimentary packages related to sea level change*". Consider 'stratigraphic concepts' to remove jargon ("sequence stratigraphy"). One could argue 'confused' in some cases.

Lines 58-60: “*thus difficult or even impossible to identify*”. I agree with this but if you highlight problems, then I urge you to make sure that the reader knows exactly what sequence stratigraphy is in the prior sentence.

Reply: We now present a concise definition of sequence stratigraphy in accordance with Catuneanu (2009) before the first appearance of the term “sequence stratigraphy”.

Comment: Lines 56-59: “*By example, sedimentary features representing sea-level fall in depositional sequences are often marked by unconformable surfaces, but may be subtler and “conformable” thus difficult or even impossible to identify*”. Rewrite.

Reply: This sentence has been changed to: “For example, sedimentary features representing sea-level fall in depositional sequences are often marked by unconformable surfaces in basin margins. Toward the basin center these unconformable surfaces may become subtle and even “conformable” thus difficult or even impossible to identify.”

Comment: Lines 59-60: “*These problems together with limited accuracy in the geologic timescale hinder the reconstruction of global sea-level and understanding the origins of sea-level change*.” Ok. This is the practical gap and your method will circumvent this.

Reply: This is our hope.

Comment: Line 65: “*The DYNOT model is consistent with the lag-1 autocorrelation coefficient, or ρ_1 model*”. consistent in what way?

Reply: “consistent with” has been changed to “supplemented by”.

Comment: Lines 67-68: “*DYNOT and ρ_1 applied to a marine slope record ...*” What exactly is measured on the marine slope to 'correlate' with the foram-derived SLC?

Reply: “marine slope record” has been changed to “marine slope gamma ray record”.

Comment: Lines 69: “*These two approaches from modeling dynamic sedimentary noise applied together with an astronomical timescale for the Early Triassic ...*” A simple explanation / statement regarding these methods is required here. Then the details follows.

Reply: We have added this statement: “This verification indicates that the sedimentary noise model is a useful method for sea-level reconstruction”.

Comment: Lines 79-80: “*water-depth related noise such as storms, tides, bioturbation, and unstable depositional rate*”. 'unsteady'?

Reply: We changed “unstable” to “unsteady”.

Comment: Line 84: “*and dating errors and depositional rate by generating an age model*”. Note that the age model is scale dependent (e.g. Sadler, 1981). So errors in dating and depositional rate are assessed against an astrologically-tuned model? How valid is this?

Reply: Advances in astrochronology enable new evaluations of depositional rate. Sadler (1981) pointed out the likelihood of larger gaps in the sedimentary record as the time span increases. This trend is significant if a time span covers multiple orders of magnitude. In our Early Triassic examples, the detected astronomical cycles are taken as a time scale. Each recognized 405-kyr cycle is the same in time duration. Astronomical tuning of cyclic stratigraphy has been shown to suppress dating errors, as demonstrated in the important publications of Kuiper et al. (Science, 2008) and Meyers et al. (Geology, 2012).

Comment: Lines 86-88: “*When sea level is high, water-depth related noise at a fixed slope location in the marginal marine environment is relatively weak; when sea-level is low, the noise is relatively strong.*” Agree. But what is 'low' and what is 'high'? What mechanism is responsible for this noise?

Reply: We have rewritten this sentence “When sea level is relatively high, water-depth related noise at a fixed slope location in the marginal marine environment is weaker than the noise in a time of relatively low sea-level, and vice versa”. For noise-generating mechanisms, we already presented potential sources in the original text: “(i) water-depth related noise such as storms, tides, bioturbation, and unsteady depositional rate.”

Comment: Lines 90-91: “*The DYNOT model is designed to measure water-depth related noise as an indicator for relative sea-level changes.*” Perhaps this will be covered later, but what calibration do you use to ensure that water-depth related noise is present? Or rather, that the noise IS water depth related? Do you have analyses of a modern environment for instance?

Reply: This sentence has been changed to “The DYNOT model is designed to measure noise in climate and sea-level proxies. If proxy-related noise and other factors, i.e., noise sources (ii) and (iii) are minor, the variance of the noise can be an indicator for relative sea-level changes.” However, the noise model needs a running window that is hundreds of kyr in scale, therefore, it is technically not possible to apply this model in a present-day slope setting. However, global sea-level changes over the past 1.4 million years do provide a test in a relatively young record that has well-established sea level changes.

Comment: Lines 93-94: “*When sea-level is high, the DYNOT ratio is relatively weak; when sea-level is low, the DYNOT ratio is relatively high.*” Accordingly. But see earlier comment(s).

Reply: We have rewritten this sentence.

Comment: Line 95: “ *ρ_1 sea-level model*”. I appreciate that you cannot have all of the details in here but this paragraph does not really say anything.

Reply: Similar to the previous “DYNOT sea-level model” in Line 89, this presents a brief lead-in for the next paragraph. We now add “see **Methods**” in the paragraph.

Comment: Lines 96-97: “*When climate change shows persistence, it tends to incorporate previous values over a range of timescales*”. Simplify to make accessible. Explain what persistence is in this context?

Reply: We rephrased as: “Climate change tends to incorporate previous values over a range of timescales; this is termed autocorrelation or persistence.”

Comment: Lines 103-105: “*the model is assumed to be valid for slope and basin environments at water depths of several meters to several hundred meters that are near or just below storm wave base*”. Why?

Reply: We have revised this to: “the model is assumed to be valid for slope and basin environments at water depths of several meters to several hundred meters that are near or just below storm wave base, where storms, tides, bioturbation, and unsteady depositional rates are expected to exert measurable influence (noise) in sedimentary records”.

Comment: Lines 108-109: “*earthquake-induced downslope movements may affect sea level change*”. The RESULTS not 'actual' sea level change.

Reply: In this paragraph, we carefully listed caveats, including earthquake induced noise that is not related to sea level change.

Comment: Line 111: “sidebands or combination tones¹⁷ that may not be removed by the model”. Cannot?

Reply: We have changed “may not” to “cannot.”

Comment: Lines 118-119: “and affect the climate persistence”. Define earlier.

Reply: Done – see reply to Comment on Lines 96-97 (above).

Comment: Line 133: “*Long-lived unrecognized gaps lead to a slight increase in DYNOT and decrease in ρ_1* ”. Compared to what timescale or process?

Reply: We have rewritten this sentence: “Long-lived (10^5 -yr scale or longer) gaps simulating unrecognized sedimentary hiatus lead to a slight increase in DYNOT and decrease in ρ_1 ”.

Comment: Lines 142-143: “*In particular, Ocean Drilling Program (ODP) Site 1119 (Fig. 2) is located 96 km east of South Island in the Canterbury Basin*”. Provides an ideal site it seems. But what point does a field site become non-ideal? If you are performing a similar analysis on much older successions (as you do later) then you never really know the water depth, dynamics of sedimentation etc.

Reply: The purpose of the late Quaternary example is to verify DYNOT and ρ_1 modeling. The ODP Site 1119 is a good example for a known water-depth, slope depositional environment, high resolution gamma ray proxy record and high-resolution astrochronology as explained in the main

text. The succession may not be used if has numerous hiatuses, or is a pelagic depositional environment. We agree that we will never really know the water depth and dynamics of sedimentation in the deep geological past; the exciting thing we can do is advance our knowledge towards the truth. We have removed “ideal” at the beginning of the paragraph.

Comment: Lines 145-147: “*which suggests that paleoclimate proxy data at that site are susceptible to increased environmental noise during lowstands*”. why does it 'suggest'? Reword. Also the term 'lowstands' is a bit of jargon. Use 'low sea level' or 'times of lowest sea level' or other.

Reply: This sentence has been changed to “which leads to an inference that paleoclimate proxy data at that site are susceptible to increased environmental noise during times of low sea level.”

Comment: Lines 149-150: “*Evidence for gravity flows is rare*”. Which is good as it introduces 'noise' into the system? So what are the ideal site characteristics that you need for the model. It might be good to explicitly state at some point.

Reply: Yes, gravity flows distort sedimentary records. We now present the characteristics for an “ideal” succession for model verification at the beginning of the paragraph.

Comment: Lines 166-167: “*which suggests that the sources of noise at Site 1119 are different from those in global sea-level changes*”. Ensure that you restate that this result is for running windows of 400 kyr, i.e. v long time frames, geological time frames.

Reply: We have added “using a 400-kyr running window” to the sentence.

Comment: Lines 187-188: “*The Daxiakou section is 700 km distant from Chaohu, yet correlates to Chaohu section (Fig. 5b-c)*”. presumably during the late-Permian/early Triassic?
“(Fig. 5b-c)”: correct figure reference?
“yet correlates to ...”: Reword. Are you suprised that they correlate? A simple statement that says that they correlate which means or 'this validates' or 'sets a framework' etc

Reply: The 700-km distance is present-day distance. Both Chaohu and Daxiakou are in the South China plate. Deformation within the South China plate in the Cenozoic and Mesozoic has been insignificant. The sentence has been changed to “The Daxiakou section is currently 700 km distant from Chaohu (similar distance in the Early Triassic)”. The “(Fig. 5b-c)” has been changed to “(compare Fig. 6b-c for Daxiakou and Fig. 6d-e for Chaohu)”. We have also removed “yet” and added one sentence “This sedimentary noise modeling sets a new framework for Early Triassic sea levels in South China.”

Comment: Lines 195-196: “*ripples and cross stratification, correlate with increased noise levels at Chaohu (Supplementary Fig. 8-9)*”. Ok - but specifically do they offer in terms of noise? What are they physically contributing to when it comes to the model?

Reply: This sentence has been revised. Both state that sedimentary structures and increased noise levels indicate shallow sea levels.

Comment: Lines 199-201: “Here, DYNOT and ρ_1 modeling provides an unprecedented, high-resolution time scale estimated directly from stratigraphy for global correlation”. How did you generated this unprecedented, 'high resolution timescale'? I'm not suggesting that you have not done this but it requires an explicit statement somewhere of where and how this was done.

Reply: The word “unprecedented” has been removed. We have also added one sentence to state how the timescale was formulated after we present the correlation between Chaohu and Daxiakou sections (previous paragraph, Line 188).

Comment: Lines 203-205: “The amplitudes of these sea-level falls, i.e., major 3rd order sequence boundaries, have been interpreted to be as much as 75 m”. Avoid sequence strat jargon and terminology.

Reply: “, i.e., major 3rd order sequence boundaries,” has been removed.

Comment: Lines 206-207: “establishing the global nature of these 10⁶-year scale eustatic events”. and synchronicity? Ok so you are laying the foundation for the next section.

Reply: We have added “and synchronicity” after “the global nature.”

Comment: Lines 208-210: “These global sea level changes are anti-phased with water storage variations recorded in terrestrial sequences deposited in lacustrine and fluvial environments of the restricted Germanic Basin”. A bit of a leap into 'water storage'. I would like to see more explanation and the evidence used to infer 'water storage variations'. “water storage variations” - how measured? “terrestrial sequences deposited in lacustrine and fluvial environments” - Right. I'm not with you. what are you measuring in these terrestrial sections?

Reply: This comment helps us fill a gap in the original manuscript. We have now added three new paragraphs on the “hypothesis of aquifer eustasy.” The linkage between “non-marine sequence” and water storage variations is also presented.

Comment: Line 211: “water storage declines in the uppermost Zechstein Group”. What is 'water storage' then? How does its decline generate sequence boundaries?

Reply: We have changed “water storage declines” to “lake levels and groundwater tables decline.”

Comment: Lines 216-218: “Orbitally-tuned magnetostratigraphic correlation between South China and Germany^{14,27} provides a high-resolution time scale and reveals that major continental water-storage falls”. “high-resolution time scale” But only as good as your orbital tuning.

Reply: “a high-resolution time scale” has been changed to “an integrated time scale”.

Comment: Lines 221-224: “Regional tectonics may have contributed to water storage changes in the Germanic Basin²⁶. However, climate simulations and sedimentological evidence for

alluvial plain and playa-lake deposits suggest groundwater fluctuations resulting from precipitation on the remnants of the Hercynian (Variscan)–Appalachian Mountains^{29,30}. I would like to see the evidence against tectonics rather than evidence for climate. What is this?

Reply: We have rewritten the sentences on tectonics.

Comment: Lines 227-228: “*water masses “see-sawed” between land and ocean in the Early Triassic hothouse (Figs 5 and 6)*”. OK, but I must be persuaded of the driver. Are there other Triassic sequences that show a similar climate-driver?

Ok - I see what you are getting at but please expand and state clearly what you mean with regard to onshore offshore sea-saw of water masses.

Reply: This section discusses the driver of aquifer eustasy. Until now, there has been no robust evidence presented of other Triassic sequences showing a similar climate driver. We have presented in a previous paragraph: “However, correlations of terrestrial and marine sequences in the Cretaceous and Late Triassic that support the aquifer eustasy hypothesis are enlightening but suffer from a lack of reliable chronology”. We also now change the “see-sawed” between land and ocean” to “see-sawed” between continental reservoirs and ocean”.

Comment: Line 229: “*Astronomical forcing of land-ocean water balance dynamics*”. Apologies if I have missed a point, but this is a bit of a step without explanation how we got here. Perhaps if you responded to my comment above and added a bit more explanation it might help. I know you stated that 'global sea level changes are anti-phased with water storage variations' but no details were offered.

Reply: We have completely rewritten the paragraphs on the global sea-level and continental water storage correlation.

Comment: Line 230: “*The mechanism for high-amplitude water exchange between continental reservoirs and the ocean*”. What is 'high amplitude water exchange'?

Reply: “high-amplitude water exchange between continental reservoirs and the ocean” has been changed to “high-amplitude glacio-eustasy”.

Comment: Line 240-242: “*An alternative model of “aquifer eustasy” or “limno-eustasy” implies that 10⁵ to 10⁶ year scale variations in continental water storage*”. So the rocks themselves store the water? This is the mechanism? So what information have you gathered from the Germanic terrestrial sequence to come to this conclusion?

Reply: We have moved this paragraph into the previous section on “Evidence for land-ocean water balance dynamics”. This paragraph has also been completely rewritten to present more details on the mechanism of the aquifer eustasy hypothesis.

Comment: Line 248-249: “*geological evidence that continental aquifers had a major impact on global sea level*”. Ok - I need to look over what data was used. Line 249-250: “*Lower water storage in the Germanic Basin interpreted from facies changes*”. How is this done? This is very

important but not explained.

Reply: We have completely rewritten the geological evidence on the “Evidence in the Early Triassic”. Detailed explanations on how water storage in the Germanic Basin is inferred are presented. The “facies changes” has been changed to “sequence stratigraphy”.

Comment: Lines 258-261: “*Thus, a variable precipitation intensity may have led to changes in aquifer capacity in the Germanic Basin. Recharging and drainage of groundwater and lakes indicates a significant water exchange between land and ocean, which may have paced the evolution of terrestrial ecosystems, e.g., spore and pollen diversity (Fig. 5k).*” What are the details of this mechanism?

Reply: We have completely rewritten these sentences. A detailed mechanism is now presented.

Comment: Line 286: “*water storage variations in the continental Germanic Basin*”. Again, be clear on how this is determined. I may have missed something, which means that others might.

Reply: The missing information on the water storage variations has been provided in the revised manuscript. The “water storage variations” has been changed to “water storage variations inferred from sequence stratigraphy”.

Response to reviewer #2: Jens Wendler

Comment:

This manuscript presents a novel model approach to robustly correlate sedimentary sequences governed by sea-level fluctuations. It aims at extracting cyclic sea-level signals and separating them from a variety of noise which previously prevented reliable correlation in sequence stratigraphy. The study benefits from a convincing test using a young record of well-established sea-level changes.

The paper then goes a significant step further to what I consider its most important outcome: It shows on the level of model-correlation that in the early Triassic - a prime study interval for non-glacial sea-level fluctuations - global ocean levels changed in anti-phase with continental lake levels. This supports the hypothesis of aquifer eustasy by suggesting, for the example of the Germanic Basin, that continental precipitation was the main factor responsible for rises and falls of this playa lake.

This paper will be of broad interest to stratigraphers, sedimentologists, paleo-climatologists etc. Importantly, it is very much needed for correct correlation attempts in sequence stratigraphy, and by providing evidence of the aquifer-eustatic process the present paper will influence thinking in this broad field of geoscience.

The manuscript is well-organized, technically well-performed, and I consider it mostly publishable at its present form.

The statistical approach of the presented new DYNOT model is robust and has been exposed to a strict Monte Carlo uncertainty analysis.

I recommend the following additional short discussion on the forcing of the Germanic Basin lake-level: The authors should address the previously postulated connecting straits between the Germanic Basin and the Tethys ocean, like the Schlesian Strait. As it is shown that global ocean-level rise was in anti-phase with the lake-level it follows, that marine ingressions through these straits cannot have been a significant contribution of water to the lake, as has been previously thought.

This paper stimulates exciting new research questions along the lines of the above thought. Obviously it should not lose conciseness by delving much into these aspects, yet I recommend placing a note on the need for research addressing these questions.

Using the attached annotated ms-file, please correct minor errors in writing and respond to the minor comments. Regarding data availability, it would be preferable if the data of this study could be stored in an online database.

Reply: We now provide additional information to discuss the connection between the Germanic Basin and the open sea. All minor errors in the annotated ms and SI files have been corrected. Point-to-point responses are listed below. The manuscript also includes links to the data that support our findings.

Comment: Line 62: “*the causes of high frequency, high-amplitude sea-level changes*”. The ~1 Myr sea-level cycle should not be called “high-frequency”! As you tackle that particular one please be clear in terminology here.

Reply: The term “high frequency” has been changed to “million-year (myr) scale”.

Comment: Lines 177-179: “*Sediments at both Chaohu and Daxiakou sections deposited in an offshore slope to basin setting in the early Early Triassic*”. Please coordinate this statement with the description of sedimentological features on page 20: the recognition of hummockies, ripples and other high energy features along with the possibility to indicate maximum flooding surfaces is not consistent with a basin setting. I would rather propose an outer shelf to slope position.

Reply: We revised the sentence to “... an offshore slope to basin setting in the early Early Triassic, and in proximal ramp to outer shelf conditions in the late Early Triassic”

Comment: Lines 212-213: “*water storage declines in the uppermost Zechstein Group, the base of the Volpriehausen, Detfurth, Solling and Röt formations*”. You should specify here, that all but the Röt – mfs are anti-phased. The Röt transgression indeed reflects major marine ingression and as such it would be expected that this transgression is in-phase with global sea level. And I can in fact see this well in fig. 5 g, h.

Reply: We have added more information on this anti-phased relationship in the late Spathian through the earliest Anisian. The Röt transgression is now also addressed. In this revision the Germanic sequences have been moved from the SI file into the main text.

Comment: Line 222: “Regional tectonics may have contributed to water storage changes in the Germanic Basin”. Include here some words on the widespread thinking that the Germanic Basin level fluctuations were related to marine ingressions. Because in that case the Germanic sequences could also be in phase with the global sea level.

I even feel that based on your precise astrochronological German-china link PLUS the observation of the anti-phase correlation you could make a point here to speak against effective connections between the Germanic B. and the world ocean.

Reply: We have incorporated these ideas and arguments into the main text.

Comment: Lines 258-259: “the 1.2-myrcycle obliquity variation maxima are linked to re-invigorated heat and moisture transportation and intensified precipitation^{35,36}”.

For substantial evidence of the link between the 1.2 Myrcycle, precipitation and sea-level please add here the following citation:

Wendler, J.E., Wendler, I., Vogt, C., Kuss, H.J., 2016. Link between cyclic eustatic sea-level change and continental weathering: evidence for aquifer-eustasy in the Cretaceous. *Palaeogeogr. Palaeoclimatol. Palaeoecol.* 430–437.

and:

Wendler, J.E., Meyers, S.R., Wendler, I., Kuss, J., 2014. A million-year-scale astronomical control on Late Cretaceous sea-level. *Newsl. Stratigr.* 47, 1–19.

Reply: We have added these key references.

Comment: Second paragraph in Section 5, page 3 in SI. Repeats part of the main ms. Is it necessary here?

Reply: Reviewer #1 had a similar comment. We have moved the entire Section 5, page 3 in the SI file into the main text.

Comment: Suppl. Fig. 8: Explain the green symbol next to the fish.

Reply: The symbol represents “reptile fossil,” and has been clarified.

Comment: Suppl. Fig. 13: Correct spelling: Middle panel: correct “ö” in Calvörde. Left/middle panel: Magnetochrons

Reply: Done.

Response to reviewer #3: Dr. Graham Weedon

Comment:

Li et al provide a novel method for estimating medium term (1 to 2 Myr) variations in sea level using analysis of time series that are proxies for sediment composition. As explained below I would support publication after minor revision.

Firstly they developed a new method for analysis demonstrated for a Pleistocene to Recent section from ODP Leg 181 off New Zealand and then applied the method to two Chinese natural gamma ray logs from the Triassic. The method simply involves removing the quasi-regular orbitally-driven components of the time series and then find the time-varying ratio of the "non-orbital" variance to the total (untreated) time series variance. As a check they have used the time-varying lag-1 autocorrelation (which is largely independent of the regular components). Their method is novel and, accepting their interpretation that the results indicate 1-2 Myr variations in sea level, potentially of great value.

It should be pointed out that they have only used one Pleistocene case of just 1.4 Myr long to test their data. Much longer sea-level related records from passive continental margins could be used for testing (e.g. the New Jersey margin investigated by the IODP Leg 313 following up earlier ODP studies). On the other hand, this idea deserves investigation so I'm happy to go along with their argument that they have recovered a long-term sea level change off New Zealand in the Pleistocene and that they have correctly interpreted the meaning of the Chinese results too.

Reply: The idea of testing the model using longer sea-level records such as those on the New Jersey margin deserves future investigation. We elected not to focus on New Jersey in this paper because (1) the New Jersey records from Leg 313 are for Miocene-Oligocene sea-level, during which reference sea levels may not be as reliable as those for the most recent 1.4 million years. In our revision we have incorporated this point (reliable reference sea levels) at the beginning of the revised section "Model verification: the late Quaternary"; and (2) a comprehensive presentation of such work will further dilute the main focus of the paper, which is Triassic evidence of aquifer eustasy. We agree that the DYNOT and ρ_1 modeling results with a 400-kyr running window makes a 1.4 myr sea level reconstruction look a little bit short. The manuscript also presents modeling results from a shorter, 100-kyr running window. Those results are presented in the SI (Suppl. Fig. 7) because the 100-kyr running window was not used in the Triassic case. A discussion of the results using a smaller running window is also presented in the **Methods**.

Secondly having judged their Chinese results to represent global sea level variations they explain the variations themselves in terms of 1-2 Myr cycles in palaeoclimate as driven by long terms of the orbital obliquity. Overall the treatment is detailed and convincing, but I did not immediately understand the key points listed above because there are some aspects of the paper that could do with clarifying. The suggestions made here are provided to help improve clarity. They do NOT represent a criticism of the arguments or of the conclusions.

Reply: Reviewer #1 had similar comments. In this revision we have added detailed information on the aquifer eustasy hypothesis, and discussion of the linkage among astronomical forcing, precipitation, filling and discharge of continental aquifers, and sea level. The time scale of these processes is also clarified.

Suggestions for improvement:

1) The title could be much clearer. The author describe the "non-orbital" variance extracted with their main method as "Dynamic sedimentary noise". This is a strange use of the word dynamic and as it is not standard I think most readers will not have a clue what the paper refers to from the title. Similarly the second part of the title refers to "land-ocean water balance dynamics". This also fails to connect the reader with the idea that 1-2 Myr sea level variation in the Triassic was driven by long-term obliquity variations. Why not change the title to something like: "Sedimentary noise and Triassic sea levels linked to long-term obliquity forcing"?

Reply: We now change the title to “Sedimentary noise and Triassic sea levels linked to land-ocean water balance dynamics and long-term obliquity forcing”. We have retained the word “dynamics” because of new explicit, expanded discussions on “land-ocean water balance dynamics” now provided in this revision.

2) The method is said to remove the orbitally forced components of the compositional variability from the time series (built into the term DYNOT). However, strictly it only removes the orbital components between 405 kyr and 18 kyr. The reasoning that Triassic (hothouse) sea level variations over 1-2 Myr are related to long-term obliquity forcing of climate means the authors think there are long-term orbital components in the DYNOT estimates. Therefore it needs to be made much clearer that the DYNOT estimates only eliminate the "short term" orbital components. To clarify this the authors should change the abstract on line 1 from "high-frequency sea-level" to "high-frequency (1/1-2 Myr) sea-level" [though I personally don't think of 1/1-2 Myr as high frequency]. Similarly on line 32 of the abstract change "astronomically forced water mass exchange" to "long-period (1-2 Myr) astronomically forced water mass exchange".

Reply: Line 21: "high-frequency sea-level" has been changed to “million-year (myr) scale sea-level”. Line 32: "astronomically forced water mass exchange" has been changed to "long-period (1-2 myr) astronomically forced water mass exchange". Note that we follow the recommendation of the GTS2012 book (Gradstein et al., 2012) to use “myr” other than “Myr” as an abbreviation for “million years”.

Other points:

The authors have been careful to list caveats from line 106 page 5. However, they have not mentioned that interpolation of irregularly spaced data (inc. natural gamma ray counts at ODP1119 put onto a time scale) leads to increased lag-1 autocorrelation/decreased DYNOT (avoidable by processing the uninterpolated data using the Lomb-Scargle transform).

Reply: We now address this in the revised main text, writing: “Interpolation of irregularly spaced data can also affect the model, for example, upsampling to increase sampling rate leads to artificially high ρ_1 values (see **Methods**).”

Page 6 line 132 "*may not detected*" > "*may not be detected*".

Reply: Fixed.

Page 8 line 162 "*Supplementary Fig. 5*" > "*S... Fig. 4*"

Reply: Fixed.

Page 8 line 166 Ditto Fig. 6" > Ditto Fig. 5)"

Reply: We re-checked the order, and found no problem here.

Page 12 line 251-252 "*between obliquity forcing and sea-level change*" > "*between long-term obliquity forcing and million-year sea-level change*".

Reply: "between obliquity forcing and sea-level change" has been changed to "between long-term obliquity forcing and myr scale sea-level change".

Page 12 line 262 "*Evidence of obliquity-forced*" > *Evidence of million-year obliquity-forced*"

Reply: "Evidence of obliquity-forced" has been changed to ""Evidence of myr scale obliquity-forced"".

Page 13 line 271 "*lower obliquity*" > "*lower average obliquity*"

Reply: In this case we refer to present-day obliquity angle decreasing, which will continue over the next 15 to 20 thousand years into the future. This is not “average” angle, but actual instantaneous angle. Note the original sentence has been moved to the end of the paper.

Page 13 line 281 "*the classic d18O*" > "*the classic low-passed d18O*".

Reply: Fixed.

Page 13 line 287 "*demonstrates dynamic water mass exchange*" > "*demonstrates long-term (>1 Myr) water mass exchange*" [how can water mass exchange NOT be dynamic??]

Reply: "demonstrates dynamic water mass exchange" was changed to "demonstrates long-term (1-2 myr) water mass exchange".

Page 14 line 308 End of sentence: add ref to 19, 35, 48 (as used on p18 line 401).

Reply: Done.

Page 16 lines 347-351 Again be careful interpolation leads to increase in lag-1 autocorrelation (noted above).

Reply: Yes, we presented this effect in the last paragraph of this section.

Page 35 line 720 "*interpolated time series*" > "*interpolated, tuned time series*".

Reply: Done.

Fig 6 Could easily be dropped altogether (the reader can easily visualize the idea of transfer of water onto and off continents affecting sea level).

Reply: We agree that geologists can imagine the transferring of water affecting sea-level with little difficulty. Non-geologists will hopefully also be reading our paper in Nature Communications. This picture clearly expresses our ideas – and there are some small surprises that are otherwise easily overlooked. With permission, we would like to keep this figure (now Figure 7).

REVIEWERS' COMMENTS:

Reviewer #1 (Remarks to the Author):

Dear authors

Thank you for your revisions. I have looked over the rebuttal document and I am happy with your replies and manuscript changes. The main issue that remains is the quality, clarity and impact of the conclusions. You have made all the major and exciting scientific findings in the manuscript up to now, but then the conclusion falls a bit flat. In my original comments I suggested that the weakness of the conclusion might be due to scientific dilution given that the manuscript has a dual goal; the method and the scientific result. Whether this is true or not does not change the fact that the conclusion is still weak in my opinion. I suggest that you keep it broad and focus on the outcomes of the science rather than the method; but this is not to say that you should not briefly mention the new and novel tool you have developed.

Sentences and spelling require checking.

Thanks, Rob

REVIEWERS' COMMENTS:

Reviewer #1 (Remarks to the Author):

Dear authors

Thank you for your revisions. I have looked over the rebuttal document and I am happy with your replies and manuscript changes. The main issue that remains is the quality, clarity and impact of the conclusions. You have made all the major and exciting scientific findings in the manuscript up to now, but then the conclusion falls a bit flat. In my original comments I suggested that the weakness of the conclusion might be due to scientific dilution given that the manuscript has a dual goal; the method and the scientific result. Whether this is true or not does not change the fact that the conclusion is still weak in my opinion. I suggest that you keep it broad and focus on the outcomes of the science rather than the method; but this is not to say that you should not briefly mention the new and novel tool you have developed.

Sentences and spelling require checking.

Thanks, Rob

Reply: We added one paragraph in the “Discussion” part. The new paragraph highlights the major and exciting scientific findings of the paper. We checked sentences and spelling.